# Chemical characterization of organic vapors from wood, straw, cow dung, and coal burning

Tiantian Wang[1], Jun Zhang[1], Houssni Lamkaddam[1], Kun Li[1, a], Ka Yuen Cheung[1], Lisa Kattner[1], Erlend Gammelsæter[1, b], Michael Bauer[1], Zachary C.J. Decker [1, c, d], Deepika Bhattu[2], Rujin Huang[3], Rob L. Modini[1], Jay G. Slowik[1], Imad El Haddad[1], Andre S. H. Prevot[1, *], David M. Bell[1, *]

1 Laboratory of Atmospheric Chemistry, Paul Scherrer Institute, Villigen, 5232, Switzerland

2 Department of Civil and Infrastructure Engineering, Indian Institute of Technology Jodhpur, 342037, India

3 Institute of Earth Environment, Chinese Academy of Sciences, Xian 710061, China

a now at: Environmental Research Institute, Shandong University, Qingdao, 266237, China

b now at: Department of Chemistry, Norwegian University of Science and Technology, Trondheim, 7491, Norway

c now at: NOAA Chemical Sciences Laboratory (CSL), Boulder, CO 80305, USA

d now at: Cooperative Institute for Research in Environmental Sciences, University of Colorado Boulder, Boulder, CO 80309, USA

Correspondence to: Andre S. H. Prevot (andre.prevot@psi.ch) and David M. Bell (david.bell@psi.ch)

## Abstract

Solid fuel (SF) combustions, including coal and biomass, are important sources of pollutants in the particle and gas phase and therefore have significant implications for air quality, climate, and human health. In this study, we systematically examined gas-phase emissions using the Vocus proton-transfer-reaction time-of-flight mass spectrometer, from a variety of solid fuels, including beech logs, spruce/pine logs, spruce/pine branches and needles, straw, cow dung, and coal. The average emission factors (EFs) for organic vapors ranged from 4.8 to 74.2 g kg$^{-1}$, depending on the combustion phases and solid fuel types. Despite slight differences in modified combustion efficiency (MCE) for some experiments, increasing EFs for organic vapors were observed with lower MCE. The relative contribution of different classes showed large similarities between the combustion phases in beech logs stove burning, relative to the large change in EFs observed. The $C_xH_yO_z$ family is the most abundant group of the organic vapor emitted from all SF combustion. However, among these SF combustions, a greater contribution of nitrogen-containing species and $C_xH_y$ families (related to polycyclic aromatic hydrocarbons) is observed in the organic vapors from cow dung burning and coal burning, respectively. Intermediate volatility organic compounds (IVOCs) constituted a significant fraction of emissions in solid fuel combustion, ranging from 12.6% to 39.3%. This was particularly notable in the combustion of spruce/pine branches and needles (39.3%) and coal (31.1%). Using the Mann-Whitney U test on the studied fuels, we identified specific potential new markers for these fuels based on the Vocus measurements. The product from pyrolysis of coniferyl-type lignin and the extract of cedar pine needle were identified as markers in the spruce/pine branches and needles open burning (e.g., $C_{10}H_{14}O_2$, $C_{11}H_{14}O_2$, $C_{10}H_{10}O_2$). The product ($C_9H_{12}O$) from the pyrolysis of beech lignin was identified as the potential new marker for beech log stove burning. Many series of nitrogen-containing homologues (e.g.,

$C_{10}H_{11-21}NO$, $C_{12}H_{11-21}N$, $C_{11}H_{11-23}NO$ and $C_{15}H_{15-31}N$) and nitrogen-containing species (e.g.,
acetonitrile, acrylonitrile, propanenitrile, methylpentanenitrile) were specifically identified in cow dung
burning emissions. Polycyclic aromatic hydrocarbons (PAHs) with 9-12 carbons were identified with
significantly higher abundance from coal burning compared to emissions from other studied fuels. The
composition of these organic vapors reflects the burned solid fuel types and can help constrain emissions
of solid fuel burning in regional models.
**Keywords**: Vocus, solid fuel, primary emission, potential markers, combustion phase

# 1 Introduction

Solid fuels (SFs), including coal and biomass, are a primary source of domestic heating worldwide (Tao
et al., 2018; Oberschelp et al., 2019; Wu et al., 2022). In developing regions, such as India, more than
80% of rural households use biomass as cooking fuel (Balakrishnan et al., 2011). Firewood is mainly
used for rural households, followed by crop residues and cow dung 'cakes', which are made of a mixture
of dried cow dung and crop residues (Loebel Roson et al., 2021; Chandramouli and General, 2011). In
Europe, fireplaces and woodstoves are used for domestic heating in winter, which have considerable
impacts on air quality, resulting in intense 'smog' events (Kalogridis et al., 2018; Fourtziou et al., 2017;
Bailey et al., 2019; Font et al., 2022). China is the largest producer and consumer of coal in the world.
In China and some Eastern European countries like Poland, coal is widely used for domestic purposes,
such as heating and cooking of households, due to its cost-effectiveness and easy accessibility (Guo et
al., 2021; Stala-Szlugaj, 2018). The combustion of these solid fuels has been recognized as the main
source of anthropogenic emission of atmospheric pollutants that elicit adverse effects on air quality and
human health (Wu et al., 2022; Zhang and Smith, 2007).
Wildfires or bushfires have become more frequent in many regions due to heatwaves and drought
(Weber and Yadav, 2020; Williams et al., 2012). SF combustion, including wildfires, is a major source
of organic vapors to the atmosphere, emitting hundreds to thousands of different organic gas-phase
species (Hatch et al., 2019; Koss et al., 2018; Permar et al., 2021). Once emitted, evaporated vapors or
freshly emitted burning organic vapors will oxidize to produce oxygenated organic vapors with a broad
volatility range. These organic vapors with sufficiently low volatility will nucleate or condense onto
pre-existing aerosols to form secondary organic aerosols (SOA) (Kumar et al., 2023).
The identification of potential markers for each emission source will be highly valuable in evaluating
SOA formation potential and ambient source contributions. Liu et al. (2008) identified potential volatile
organic markers for different emission sources (e.g., biomass burning (BB), mobile sources and
painting). Nevertheless, these commonly used potential markers are well-established, yet due to their
presence in more than one type of biomass fuel, distinguishing between different biomass-burning
sources presents challenges. Since 2009, there have been many advancements in the gas-phase
measurements of SF, which include lab studies (Bruns et al., 2017; Bruns et al., 2016; Bhattu et al.,
2019) and large field campaigns (e.g., WE-CAN Aircraft Measurements, FIREX-AQ campaign)
(Permar et al., 2021; Jin et al., 2023; Majluf et al., 2022). However, efforts toward understanding SOA
formation in burning plumes have been hindered by limited identification and quantification of organic
vapors emitted by fires, especially intermediate volatility organic compounds (IVOCs) (Akagi et al.,
2011). Laboratory and field campaigns suggest that intermediate volatility organic compounds are
important precursors of SOA. Grieshop et al. (2009) demonstrated that traditional SOA precursors
account for less than 20% of the observed SOA formed from residential wood combustion emissions,
while IVOCs can contribute approximately 70% of the formed SOA (Li et al., 2024), which highlights
the urgent need for more research on IVOCs from BB emissions. Adding an IVOC emission inventory
to an air quality model can significantly narrow the gap between the estimated and measured SOA
concentrations(Li et al., 2024; Hodzic et al., 2010; Zhao et al., 2016; Robinson et al., 2007).
Offline sampling methods such as canisters and adsorption-thermal desorption (ATD) cartridges, along
with gas chromatography (GC) analysis, have limitations related to their low time resolution,
susceptibility to sampling artifacts, and a limited range of measurable compounds (Hatch et al., 2018;
Hatch et al., 2017). In addition to offline techniques, proton-transfer-reaction mass spectrometry (PTR-
MS) has been widely used for the online measurement of volatile organic compounds (VOCs) in the
atmosphere (Yuan et al., 2017). However, IVOCs still suffer from high losses in the sampling lines and
PTR-MS drift tubes. Furthermore, most studies have focused on either primary or aged emissions, with
very few examining the real-time influence of combustion conditions on the composition of emitted
organic vapors (Bruns et al., 2016; Akherati et al., 2020; Tkacik et al., 2017). The recently developed
Vocus PTR-TOF (hereafter Vocus) has greatly enhanced sensitivity due to a newly designed chemical
ionization source (Krechmer et al., 2018), and it can detect a broader spectrum of VOCs, IVOCs, and
their oxygenated products (up to six to eight oxygen atoms for monoterpene oxidation products) (Li et
al., 2020; Wang et al., 2021; Riva et al., 2019). With a novel design and chemical ionization source, the
Vocus allows for real-time characterization of gas-phase emissions during various burning phases (e.g.,
flaming and non-flaming phases) and identifies the potential markers for a wide range of fuels.
The present study compares real-time emissions from different combustion fuels. We begin by
demonstrating the evolution of gas-phase emissions during burning cycles highlight the changes in the
composition of the emissions. Then, we systematically characterize the organic vapors composition
using Vocus from a variety of burning fuels from both residential stoves (beech logs, spruce/pine logs,
and coal) and open combustion (spruce/pine branches and needles, straw, cow dung). We evaluate the
potential markers and EFs for different fuels and explore the dependence of individual organic vapor
emission intensity, variability, and chemical composition on solid fuel types and combustion phases.
We also discuss potential markers for the burning fuels examined in this study. The potential markers
are identified as statistical outliers determined with a Mann-Whitney test, consistent with previous
measurements (Zhang et al., 2023). The differences in EFs and profiles between different combustibles
can be considerable, and these results illustrate the importance of considering these emission sources
individually. Measurements capable of identifying and quantifying rarely measured and presently
unidentified emissions of organic vapors, particularly chemically complex SVOCs and IVOCs, are vital
for advancing the current understanding of the impact of solid fuel combustion on air quality and climate.

## 2 Materials and methods

### 2.1 Fuel and burning types

The experiments were conducted at the Paul Scherrer Institute (PSI) in Villigen, Switzerland. The
burning facility is part of the PSI Atmospheric Chemistry Simulation chambers (PACS). Real-time
characterization of the primary gas and particle phase emissions was carried out during 28 test burns.
Six solid fuels were studied (coal briquettes and biomass fuels: beech logs, spruce/pine logs, fresh
spruce/pine branches and needles, dry straw, cow dung) with three to six replicate burns. Material in
the beech, spruce, and pine fuels (e.g., logs and needles) was sourced from a local forestry company in
Würenlingen, Switzerland. Cow dung cakes (a mixture of cow dung and straw) were collected from
Goyla Dairy in Delhi, India. Coal briquettes were purchased from Gansu, China (Ni et al., 2021; Klein
et al., 2018).
With those six different fuels, we categorized six burning types for this experiment. 1) beech logs stove,
2) spruce/pine logs stove, 3) spruce/pine branches and needles open, 4) dry straw open, 5) cow dung
open and 6) coal stove. We selected these six solid fuels and conducted emissions tests to simulate
certain types of burning found in the atmosphere. Among the list above, 1) beech logs stove and 2)
spruce/pine logs stove are representative of residential wood burning, which are burned separately in a
stove, consistent with the materials used in two previous articles (Bertrand et al., 2017; Bhattu et al.,
2019). To represent forest fires or wildfire and agricultural field combustion, 3) a mixture of fresh
spruce/pine branches and needles and 4) straw were combusted in an open stainless-steel cylinder (65
cm in diameter and 35 cm in height). Traditional cooking and heating practices in regions like India are
represented by 5) cow dung cakes open burning by using half-open stoves (Loebel Roson et al., 2021).
Finally, traditional cooking and heating practices in rural regions of developing countries are
represented by 6) coal stove burning in a portable cast iron stove purchased from the local market (Liu
et al., 2017). Of course, these conditions do not fully accurately represent the conditions found in actual
fires, which consistent of a variety of burning species (e.g., trees, underbrush, peat soils, etc.…), but
represent laboratory burning conditions.

## 2.2 Experimental setup and instrumentation

The experimental design is shown in Figure S1. In summary, it is made up of a burner and a set of
diluters with heated lines. The zero air was provided by a zero air generator (737-250 series, AADCO
Instruments, Inc., USA) for cleaning and dilution (Heringa et al., 2011; Bruns et al., 2015). The zero air
generator takes ambient air and scrubs particulates and volatile organic compounds from the air leaving
a mixture that is largely made up of $N_2$, $O_2$, and Ar at ambient concentrations. Other trace gases are
scrubbed to lower than atmospheric concentrations including $CO_2$ (< 80 ppb) and $CH_4$ (< 40 ppb).
Before each burn, a continuous stream of zero air was passed through the gas lines overnight to avoid
cross-contamination between burns and to ensure a low background of VOCs. Once a burn is initiated
from the various combustibles, emissions are sampled from the chimney through a heated line (473 K).
The emissions (both gas and particle phases) are then diluted by two Dekati diluters (DI-1000, Dekati
Ltd.) which dilutes the emissions by a factor of ∼ 100 (473 K, DI-1000, Dekati Ltd.). Note that beech
logs combustion cycles consist of a first cycle referred to as the 'first load' and subsequent cycles,
referred to as 'reloads'. The first load consisted of a cold start, flaming, smoldering, and burn-out phase,
and the reloads were comprised of a warm start, flaming, smoldering, and burn-out phase. Organic
vapor emissions of solid fuel combustion are released within 10-30 min after loading according to the
properties of the fuels. We define the time until full ignition duration for burning encompasses 80% of
the entire process, starting from loading the fuels to burnout.
Numerous instruments were connected after the second dekati diluter for the characterization of both
the particulate and gaseous phases. A Scanning Mobility Particle Sizer (SMPS, CPC 3022, TSI, and

custom-built DMA) provided particle number size distribution information and calibrated by using polystyrene latex (PSL) particle size standards (Wiedensohler et al., 2018; Sarangi et al., 2017). The non-refractory particle composition was monitored by a high-resolution time-of-flight aerosol mass spectrometer (HR-ToF-AMS, Aerodyne Research Inc.). AMS data were processed using SQUIRREL (SeQUential Igor data RetRiEvaL v. 1.63; D. Sueper, University of Colorado, Boulder, CO, USA) and PIKA (Peak Integration and Key Analysis v. 1.23) to obtain mass spectra of identified ions in the *m/z* range of 12 to 120. OC (organic carbon) is derived from the ratio of organic mass (OM) to OC (OM/OC) determined with high-resolution AMS analysis (Canagaratna et al., 2015). In the AMS mass spectra, the fraction of *m/z* 60 (*f*60) represents the ratio of levoglucosan-like species (Schneider et al., 2006; Alfarra et al., 2007). AMS was calibrated for ionization efficiency (IE) by a mass-based method using $NH_4NO_3$ particles(Tong et al., 2021). Black carbon (BC) was measured with an Aethalometer (Magee Scientific Aethalometer model AE33) (Drinovec et al., 2015) with a time resolution of 1 minute. The maintenance and calibration are given in the AE33 user manual - version 1.57. A LI-7000 $CO_2$ analyzer (LI-COR) and APMA-370 CO analyzer (Horiba) provided continuous measurements of carbon dioxide ($CO_2$) and carbon monoxide (CO), respectively. The concentrations of total hydrocarbons (THC) and methane ($CH_4$) were monitored using a flame ionization detector monitor (THC monitor Horiba APHA-370).

We deployed a Vocus to measure organic vapors with a wider range of volatilities. A detailed description of the Vocus is provided elsewhere (Huang et al., 2021; Krechmer et al., 2018). For this study, the Vocus was operated with $H_3O^+$ as the reagent ion. The sample air was drawn in through a 1 m long polytetrafluoroethylene (PTFE) tube (6 mm o.d.) using a total sample flow of 4.3 L/min, which helped reduce the losses in the inlet wall and the sampling delay. Of the total sample flow, only 100-150 $cm^3$/min went to Vocus, and the rest was exhausted. The Vocus was calibrated before and after measurements every day using a multi-component standard cylinder (Tofwerk AG). Standard gases were diluted by the injection of zero air, producing mixing ratios of VOCs of around 20 ppbv. The calibration components were methanol, acetaldehyde, acetonitrile, acetone, acrylonitrile, isoprene, methyl ethyl ketone, benzene, toluene, m-xylene, α-pinene and 1,2,4-trimethylbenzene. The background measurements were performed using dry zero air every day. Data were recorded with a time resolution of 1 s. The raw data were processed using Tofware v3.2.3 software (TOFWERK, Aerodyne, Inc.). The standard non-targeted analysis workflow developed by Tofwerk was adopted for mass calibration and peak fitting. The mass transmission function and the ratios between the measured and calculated sensitivities for a series of ions were used to quantify the data and convert the ion counts to ppbv. To calculate the mixing ratio for compounds do not present in the calibration mixture, the slope of the linear fit was multiplied by the proton transfer rate constants ($k_{ptr}$) which have been provided in Supplement Table.

**2.3 Data analysis**

Modified combustion efficiency (MCE, Equation 1) is an estimate of the relative amount of flaming and smoldering and is equal to

$$MCE = \frac{\Delta CO_2}{\Delta CO + \Delta CO_2}$$
Equation (1)

Where $\Delta CO$, $\Delta CO_2$ are the mixing ratios of CO or $CO_2$ in excess of background (measured before the
combustion), respectively (Christian et al., 2003). Generally, a higher MCE ($> 0.9$) suggests dominated
flaming combustion, whereas a lower MCE ($< 0.9$) is mostly associated with smoldering combustion
(Zhao et al., 2021; Zhang et al., 2022).
The emission factors (EFs, g kg$^{-1}$) of species $i$ was calculated, following a carbon-mass balance
approach (Andreae, 2019; Boubel et al., 1969; Nelson, 1982):

$$EF_i = \frac{m_i}{\Delta mCO + \Delta mCO_2 + \Delta mCH_4 + \Delta mNMOGs + \Delta mOC + \Delta mBC} \times \cdot W_C \qquad \text{Equation (2)}$$

Here $m_i$ refers to the mass concentration of species $i$. $\Delta mCO$, $\Delta mCO_2$, $\Delta mCH_4$, $\Delta mNMOGs$, $\Delta mOC$,
and $\Delta mBC$ are the background-corrected carbon mass concentrations of carbon-containing species in
the flue gas. $W_C$ is the carbon mass fraction of the burning fuel. The $W_C$ in the fuel a constant average
value of 0.46 for wood (Bertrand et al., 2017), 0.45 for straw (Li et al., 2007), 0.45 for cow dung (Font-
Palma, 2019), and 0.49 for coal (Zhang et al., 2000) was assumed. Changes of $W_C$ over the burning
cycle are expected to be small compared to the variability of pollutant emissions. The volatility (i.e. the
saturation mass concentration, $C^*$) for individual organic compounds was calculated based on the
number of oxygen, carbon, and nitrogen atoms in the compound using the approach by Li et al. (2016):
$$log_{10}C^* = (n_C^0 - n_C^i)b_C - n_O^i b_O - 2\frac{n_C^i n_O^i}{n_C^i + n_O^i}b_{CO} - n_N^i b_N \qquad \text{Equation (3)}$$
where $n_C^0$ is the reference carbon number; $n_C^i$, $n_O^i$ and $n_N^i$ denote the numbers of carbon, oxygen, and
nitrogen, respectively, in the compound. $b_C$, $b_O$ and $b_N$ are the contributions of each atom to $log_{10}C^*$,
respectively; and $b_{CO}$ is the carbon-oxygen nonideality. The parameters used in this analysis are
presented in Table S1. Most notably, the empirical approach used by Li et al. (2016) was derived with
only a limited number of organonitrates, which could potentially introduce bias in estimating vapor
pressure (Isaacman-Vanwertz and Aumont, 2021). To mitigate this bias, we modified the nitrogen
coefficient for CHON formulas that can be forced to equal twice the negative of the oxygen atom ($b_N$
$= -2b_O$).

## 2.4 Identification of potential markers

In this study, the relative contribution of the mixing ratio for over 1,500 species from six different fuels
was quantified across all 28 test burns using the Vocus. To identify the potential markers of emissions
from different fuels, we implemented the Mann-Whitney U test (Mann and Whitney, 1947; Wilcoxon,
1945) in MATLAB®. Mann-Whitney is a non-parametric test, which has been applied in the selection
of aerosol markers (Zhang et al., 2023), proteomic markers (White et al., 2019; Chen et al., 2012;
Teunissen et al., 2011; Chmaj-Wierzchowska et al., 2015; Nomura et al., 2004), and other biomarkers
(including measurements with a PTR-MS) (Jasperse et al., 2007; Nagai et al., 2020; Sun et al., 2019;
Tritten et al., 2013). It is a nonparametric test and is used for between-group comparisons when the
dependent variable is ordinal or continuous and not assumed to follow a normal distribution with small
sample sizes. This test takes two data samples as parameters, uses the ranks as a measure of central
tendency, and then returns the test results with a $p$-value to indicate the statistical significance. When
the $p$-value is lower than the significance level of 0.1 (a commonly used $p$-value to study statistical
significance in atmospheric research), the median of the tested sample is significantly high or low in
the two-tailed test. The molecules from a specific class of fuel that satisfy the pairwise comparison test
between one fuel, referred to as fuel $j$, and other types of fuel, were determined to be significantly high-
or low-fraction ions in fuel $j$. These ions have the potential to serve as potential markers for fuel $j$. We
have calculated in addition the fold change ($FC$) of ion $i$ in fuel $j$ was calculated using Equation 4,

$$FC_{i,j} = \frac{f_{i,j}}{f_{i,other}}$$
        Equation (4)

Where $f_{i,j}$ represents the fraction of ion $i$ in the mass spectra profiles of fuel $j$, and $f_{i,other}$ represents
the average fraction of ion $i$ in the mass spectra from the other fuels.
To identify potential markers the Mann-Whitney U test was used to compare the emissions observed
for one type of fuel, (e.g., spruce/pine logs) with the gaseous emissions observed for other fuels. The
data used for the comparison was the average composition measured throughout a full burning cycle,
excluding the initial ignition period. However, due to the similarity in solid fuel types between burning
spruce/pine logs, as well as spruce/pine branches and needles, they were categorized as separate solid
fuel types for this test and not compared with each other but were only compared with the other four
types of fuels.  This could result in the loss of many same markers since these two types of fuel actually
come from the same type of tree. Therefore, when identifying markers for spruce/pine logs using the
Mann-Whitney U test, spruce/pine branches and needles were not included in the comparison group.
Similarly, due to the composition of cow dung 'cakes,' which are a mixture of dried cow dung and crop
residues, the approach used in the Mann-Whitney U test is consistent with the above method.

## 3 Results and discussion

### 3.1 The characteristics of EF and MCE from different solid fuel types

The average EFs of CO, $CO_2$, organic vapors and particular matter (PM) in g/kg as well as the MCE
values calculated for the 6 types of fuels, are shown in Table 1. Detailed EFs and MCE values for each
experiment can be found in Table S2. The average MCE value depends on the solid fuel type and the
combustion phase (flaming and smoldering) that is occurring. The lowest MCE values, 0.90, were
observed during the smoldering phase of the stove-burning of beech logs, while the highest values (0.99)
were recorded during the flaming phase of the spruce/pine branches and needles open burning. In all
experiments, the highest EFs for a single gas-phase species correspond to $CO_2$ (1136.2-1711.7 g/kg).
Coal burning has the highest average CO EFs (40.6 ± 12.6 g kg$^{-1}$) and $CO_2$ EFs (1680.2 ± 32.7g kg$^{-1}$).
Total organic vapor EFs reported in Table 1 refer to species quantified using the Vocus. The average
EFs of organic vapors (in the range of 4.8 to 74.2 g kg$^{-1}$) and the standard deviation are calculated based
on the average EFs for the repeatable experiments, which depend on the combustion phases and solid
fuel types. Generally, lower MCE values correspond to higher organic vapor EFs within a given class
of burning fuel (Figure S2a). For instance, smoldering beech logs resulted in significantly higher
average organic vapor EFs (74.2 ± 42.9 g kg$^{-1}$) compared to burning spruce/pine logs. Spruce/pine stove
and open burning, dominated by the flaming phase (average MCE > 0.95), exhibited average organic
vapor EFs of 44.9 ± 17.5 g kg$^{-1}$ and 39.8 ± 11.4 g kg$^{-1}$, respectively. This value is higher than pervious
study (37.3 g kg$^{-1}$) even though the difference is in the uncertainty levels , which can be attributed to
the more extensive analysis of organic vapor in our study (Hatch et al., 2017). Despite the slight
difference in MCE for some experiments, the increasing EFs for organic vapors with at least six carbon
atoms per molecule ($\geqslant$ C6) as proxy SOA precursors were observed with lower MCE (Figure S2b)
(Bruns et al., 2016). Moreover, the EFs of these SOA precursors are much higher than the primary
biomass-burning organic aerosol (BBOA), which suggests a higher potential for SOA formation.
Notably, the EFs of organic vapors from cow dung and coal was relatively low, at $4.8 \pm 0.98$ g kg$^{-1}$ and
$11.5 \pm 2.6$ g kg$^{-1}$, respectively. Our EFs align well with previously reported volatile organic compound
EFs from bituminous coal combustion under similar conditions (range of 1.5 to 14.1 g kg$^{-1}$) reported by
Klein et al. (2018).

## 279   **3.2 Comparison between flaming and smoldering of wood burning**

Figure 1a shows a typical burning cycle during beech log wood experiments with distinct emission
characteristics between flaming and smoldering phases. In the top panel, the MCE, CO, and CO$_2$
concentrations, along with our experimental records, are used to indicate the flaming and smoldering
stages. The flaming phase shows considerable BC emission, while the smoldering phase is dominated
by organic aerosol emissions without visible flame. The absorption Ångström exponent (AAE) during
the smoldering phase is approximately twice that of the flaming phase, possibly due to the presence of
"brown carbon" in organic aerosols. *f*60 represents the prevalence of primary combustion products such
as levoglucosan and is used as an indicator for fresh BB emissions (Schneider et al., 2006; Alfarra et
al., 2007). During the starting/flaming phase, when the temperature is higher, *f*60 increases. Whereas
for lower temperatures in the smoldering phase, *f*60 decreases (Weimer et al., 2008). The mixing ratio
of most of the compounds correlates negatively with the MCE as expected with a significant increase
in the smoldering phase (Figure 1a and Figure S3). However, some compounds like benzene have
different enhancement rates from flaming to smoldering, which is similar to previous studies (Warneke
et al., 2011).
Figure 1b illustrates the measured EFs for flaming and smoldering wood fire stages. On average, EFs
for organic vapors in the flaming stage are approximately four times lower ($31.4 \pm 7.1$ g kg$^{-1}$) than those
in the smoldering stage fires ($121.9 \pm 24$ g kg$^{-1}$). Despite significant variability in the strength of organic
vapors emissions (EFs), the average carbon and oxygen distribution of organic vapors remained largely
consistent across the combustion phases (Figure S4). Hardwood (beech) is a fibrous substance primarily
composed of three chemical elements: carbon, hydrogen, and oxygen and these basic elements are
incorporated into several organic compounds, i.e. cellulose, hemicellulose, lignin, and extractives
formed into a cellular structure (Asif, 2009). The flaming stage is associated with more complete
oxidation with a relatively higher contribution of oxygenated VOCs (OVOCs, e.g., furan, oxygenated
aromatics, O-containing, Figure S5). Conversely, during the smoldering stage, more CO and organic
vapors are emitted relative to the flaming stage (Figure 1a). OVOCs, such as carbonyl, furan,
oxygenated aromatics, and O-containing species, form the major fraction (> 88%) of emissions in both
flaming and smoldering fires. They are followed by the sum of $C_xH_y$, and SRA (5-10%). As shown in
Figure S6, the volatility distribution of the emissions between the flaming phase and smoldering phase
changes slightly with a decrease in the IVOCs from 25.8% (flaming) to 20.2% (smoldering). Though,
in absolute terms all emissions are enhanced during the smoldering phase, including IVOCs, due to the
increased EFs during the smoldering phase. As Figure 1b shows on a relative scale that there is a higher
contribution of single ring aromatics and $C_xH_y$ in the smoldering phase than flaming phase. Within these
measurements in our residential stove, we observe relatively small differences in the composition
relative to the large increase in EFs when moving from flaming to smoldering conditions.

**3.3 The characteristics of organic vapor from different solid fuel types**

**3.3.1 Chemical composition of organic vapor from combustion**

To assess the feasibility of distinguishing differences between combustion solid fuel types based on the
measured species, we evaluated the similarity of the mass spectra obtained from each experiment using
the correlation coefficient ($r$), as shown in Figure 2, organic vapors from the same burning fuel are
strongly correlated (0.82-0.99), indicating the general repeatability of the experiments. Furthermore,
we observed a weak intra-fuel correlation between coal and other biomass sources (0.44-0.78),
suggesting significant differences in chemical composition. By contrast, the separation between
different solid fuel type is not stark and all possess a correlation between 0.6-0.98. Overall, the
correlation coefficient highlights similarities between all biomass-based emissions, which will now be
discussed in detail.
Figure 2 also shows the average mixing ratio contribution of full ignition duration from $m/z$ 40 to 300
for each experiment, and is categorized into $C_xH_y$, $C_xH_yO_z$, $C_xH_yN$ and $C_xH_yO_zN$ families based on their
elemental composition. In all organic vapors, the $C_xH_yO_z$ family is the most abundant group, making
the largest contribution to beech logs (88.6%), spruce/pines logs (82.1%) and straw (81.7%). These
percentages are higher than those for coal (63.1%) and cow dung (68.9%). Coal burning results in
considerably higher contributions in the $C_xH_y$ families (33.7%) than beech logs (9.3%), consistent with
the bulk chemical composition of the fuels.
Figure 3 separates emitted compounds based on their carbon (x-axis) numbers. The dominant signals
in organic vapors for all fuels are attributed to C3-6 compounds, while more species with higher carbon
numbers (#C > 10) are observed in spruce/pine branches and needles open burning. The bin containing
Hydrogen to carbon ratios (H/C, calculated as the ratio of hydrogen atoms to carbon atoms in a
molecules) between 1.2 and 1.5 has the largest contribution in all fuels except the straw, ranging from
27% to 31.2%. Oxygen to carbon ratios (O/C, calculated as the ratio of oxygen atoms to carbon atoms
in a molecule) less than 0.15 contribute significantly to coal burning (42%), which corresponds to the
high proportion of $C_xH_y$ families (Figure 2). Wood and straw burning emitted more oxygenated organic
vapors than coal and cow dung burning with more contribution of higher O/C species (O/C> 0.5). The
results show similarities to the comparison between burning wood and cow dung in the particle phase
(Zhang et al., 2023). Specifically, cow dung exhibits a lower fraction of high O/C (0.22) compared to
other fuels studied.
We categorized organic vapors by functional groups into 10 classes based on the classifications used in
Bhattu et al. (2019). These classes include: alcohols, carbonyls (including acid), hydrocarbons, furans,
N-containing compounds, O-containing < 6 (where the number of carbon atoms is less than 6), O-
containing ≥ 6 (where the number of carbon atoms is equal or greater than 6), oxygenated aromatics,
polycyclic aromatic hydrocarbons (PAHs), single-ring aromatics (SRA). Figure S7 and Figure S8 show
a comparison of the organic vapor composition observed from different solid fuel types. The measured
emissions exhibit significantly different compositions, reflecting the variability of organic components
produced from different solid fuel types. The emissions of all solid fuels are overwhelmingly dominated

by carbonyls in the range of 23.1% (coal) to 45.1% (straw). For all emissions, furans represent the second largest group, which account for more than 14% of the emissions. Comparatively, aromatic compounds are less significant in BB: 5.9% - 12% for oxygenated aromatics, 0.5% - 2.1% for PAHs, and 2.1% - 5.8% for SRA. In contrast, aromatic emissions are relatively larger in coal burning emissions (13.6%, 8.1%, and 13.8%, respectively). Also, we note a specific difference in the oxygenated aromatic compounds and those with C > 6 for open wood burning conditions, compared to the stove. This difference may be driven by the difference in the water content of the wood, which is significantly higher for open wood burning (30-40%) compared to stove burning (10-12%). The increase in these oxygenated components comes at the expense of species containing carbonyl and furan functionalities.

Generally, the total fraction of nitrogen-containing species ($C_xH_yN$ and $C_xH_yO_zN$) is significantly higher in the organic vapors emitted from open burning of cow dung (18.8%) compared to the other fuels (2.1% to 7.3%). This trend is consistent with both our results from aerosol composition measurement and previous literature (Stewart et al., 2021b; Zhang et al., 2023; Loebel Roson et al., 2021). Generally, nitrogen containing compounds in cow dung consist mainly of one nitrogen atom and have a wide range of carbon numbers between 2 and 7 (Figure 3). Stewart et al. (2021a) also reported that cow dung was the largest emitter of nitrogen-containing organic vapors than other fuelwood and crops in India, releasing large amounts of acetonitrile and nitriles. These nitrogen-containing organic vapors are likely formed from the volatilization and decomposition of nitrogen-containing compounds within the cow dung cake, such as free amino acids, pyrroline, pyridine, and chlorophyll (Ren and Zhao, 2015; Burling et al., 2010).

### 3.2.2 Volatility of organic compounds

The parameterization described in Sect. 2.4 uses the modified approach of Li et al. (2016) to estimate the volatility of each of the measured compounds by the VOCUS in $log_{10}(C*)$ [µg m$^{-3}$]. The gaseous organic compounds were grouped into a 14-bin volatility basis set (VBS) (Donahue et al., 2006) (Figure 4). Following the suggestions in recent papers (Wang et al., 2024; Li et al., 2023; Donahue et al., 2012; Huang et al., 2021; Schervish and Donahue, 2020), the volatility was aggregated into four main classes with units of µg m$^{-3}$: VOCs as $log_{10}(C*) > 6.5$, IVOCs as $log_{10}(C*)$ between 6.5 to 2.5, semi-VOCs (SVOCs) as $log_{10}(C*)$ between 2.5 to - 0.5 and low-VOCs (LVOCs) as $log_{10}(C*) < - 0.5$).

Comparison and compilation of organic vapors sorted by volatility and functional group classification are shown in Figure 4, and the distribution of average EFs as a function of binned saturation vapor concentration is shown. The VOC class was found to be the most abundant, ranging from 58.7% to 87% (Figure S9). For all burning types, carbonyls, furans, and SRA families are overwhelmingly dominant in VOCs, accounting for more than 60% of the VOC emissions. The high fraction of oxygenated VOCs like carbonyls in BB emissions is in stark contrast to VOCs emitted from coal combustion, which is dominated by aromatic hydrocarbon emissions, particularly PAHs. This difference may be attributed to the condensed structure of coal, and lack of oxygen within the fuel itself. PAHs are a group of organic matter compounds containing multiple aromatic rings that mainly result from incomplete combustion (Mastral and Callen, 2000).

IVOCs also constituted a considerable fraction in solid-fuel combustions (from 12.6% to 39.3%), particularly in spruce/pine branches and needles (39.3%), cow dung (24.3%) and coal (31.1%) (Figure S9). Significant differences in the bulk volatility of organic compounds were observed among different

types of wood burning. In general, spruce/pine branches and needles open burning released a higher
proportion of IVOCs (39.3%) into the gas phase compared to stove logs burning (12.6% and 23.9%).
This difference may be attributed to a lower percentage of terpenes in woody tissues compared to
needle/leaf tissues (Greenberg et al., 2006). In addition, open burning wood has both a significantly
larger water content and oxygen content than stove burning, which enhances the formation of partially
oxidized organic compounds. Within the open burning experiments, the oxygenated molecules (both
aromatics and $C \geq 6$) are enhanced relative to the other experiments and result in the largest EF of
IVOCs. In addition to the burning conditions, the fuel properties are also an important factor affecting
the IVOC component. Notably, cow dung comprised a higher fraction of N-containing species within
their IVOC emissions compared to other fuels. The emissions of IVOCs characterized and quantified
in this study are important for the estimating and modeling of aged emissions and their propensity to be
able to form secondary organic aerosol.
**3.4 Chemical characteristics of dominant compounds from all biomass fuels and**
**identification of potential markers for specific solid fuels**
**3.4.1 Chemical characteristics of dominant compounds from all biomass fuels**
To conduct a comprehensive analysis aimed at identifying potential markers among emissions, the
Mann-Whitney U test (refer to Sect. 2.5) was performed on the relative contribution of primary organic
vapors derived from various fuels as measured by the Vocus. The results of the pairwise Mann-Whitney
test are presented in Figure S10, illustrating the average $-log_{10}$ $p$-value as a function of the $log_2$ fold
change ($FC$). Species that yield $p$-values lower than 0.1 in the two-tailed test for all pairwise
comparisons are deemed significantly more abundant or scarce in a particular solid fuel type compared
to all other fuels. These species are indicated as colored circles in Figure 5. In cases where species do
not meet this criterion once or multiple times, they are represented as gray circles, even if their average
$p$-value falls below 0.1. A higher $-log_{10}$ ($p$-value) signifies a reduced likelihood that the fractional
medians of two species are equivalent. Simultaneously, a greater $FC$ (as per Equation 4) indicates an
increased presence of the species' fractional contribution in the tested fuel in comparison to the average
contribution across all other fuels. This suggests a higher degree of exclusivity for this species in the
given context. The potential markers, $p$-values, fold changes, and threshold results are listed in the
Supplement Table.
As shown in Figure 2, biomass fuels (such as logs, branches, needles, straw, and cow dung) were
analyzed separately from coal due to their distinct characteristics. To address this distinction, we
characterized the dominant compounds across various biomass fuels by setting a threshold (relative
mixing ratio contribution $\geq 0.1\%$) for compounds that are not potential markers of one specific biomass
fuels. This approach allowed us to identify compounds that are more readily detectable in complex
environments. As shown in Figure S11, the gas-phase analysis revealed several dominant compounds:
$C_5H_4O_2$ (furfural, 2.2-10.1%), $C_2H_4O_2$ (acetic acid, 2.1-5.8%), $C_3H_6O_2$ (methyl acetate, 1.7-4.6%), and
$C_2H_4O$ (acetaldehyde, 1.3-3.9%), which were also reported prior studies on BB (Bruns et al., 2017;
Stockwell et al., 2015; Christian, 2004; Sarkar et al., 2016). Furthermore, the category of dominant
compounds represents the primary set of compounds associated with BB, contributing from 46% to 69%
of the emissions (Figure S12). Carter et al. (2022) expand the representation of fire organic vapors in a
global chemical transport model, GEOS-Chem, which contributes substantially to atmospheric
reactivity, both locally and globally. Our results could provide more input information for global or
regional chemistry transport models.

## 3.4.2 Identification of potential markers for specific solid fuels

Mass defect plots of potential markers are visualized in Figure 5, for each burning source, respectively.
Many potential markers are identified for each unique type of burning (Supplement Table). As shown
in Figure 5, potential markers of all wood burning are mainly composed of compounds from the $C_xH_y$
and $C_xH_yO_z$-family. However, the potential markers for spruce/pine branches and needles have higher
molecular weights and are more oxidized, which aligns with their characteristics of the mass spectrum.
In contrast, compounds from open burning of straw and cow dung contribute considerably more to
nitrogen-containing families but less to oxygen-containing species, consistent with their bulk chemical
composition characteristics. Additionally, potential markers for coal consist mainly of compounds from
$C_xH_y$-family, which also aligns with its bulk chemical composition and relatively higher H/C ratios
(Figure 3).
For all softwood (i.e., spruce/pine logs and spruce/pine branches and needles in this study),
monoterpenes ($C_{10}H_{16}$) are a potential marker along with the fragment at $m/z$ 81.07 ($C_6H_8$). However,
monoterpenes cannot exclusively be related to BB given their abundance in the atmosphere.
Monoterpenes are also the biogenic volatile organic compounds (BVOCs) emitted from natural trees
and other vegetation (Hellén et al., 2012). However, the emission rates of terpenes vary with season,
with a higher rate in spring and summer and a lower rate in autumn and winter (He et al., 2000; Noe et
al., 2012). In winter, monoterpenes could be a potential marker for softwood burning due to minor
natural emissions from spruce, but in summer, terpene emissions from natural trees would dominate the
contribution making it a non-potential marker. P-cumenol ($C_9H_{12}O$), as one of the potential markers for
beech logs, was discovered to be one of the prominent products of beech wood pyrolysis of lignins
(Sengpiel et al., 2019; Keller et al., 2020). Homologues of $C_{10}H_{8-18}O_2$ are determined for spruce/pine
branches and needles, with $C_{10}H_{10}O_2$ being β-phenylacrylic acid, which is one of the main chemical
compositions of the extract of the cedar pine needle. $C_{10}H_{14}O_2$ could be 1-guaiacylpropane, which is
proposed as a potential marker for coniferyl-type lignin pyrolysis products from pine (Simoneit et al.,
1993; Liu et al., 2021). Homologues of $C_{11}H_{8-18}O_2$ are also seen, for example, $C_{11}H_{14}O_2$, likely 1-(3,4-
dimethoxy-phenyl)-1-propene, which is stated as a representative compound found in lignin (Alves et
al., 2003; Hill Bembenic, 2011).
Noticeably, cow dung has a significantly different chemical composition. As a result, many potential
markers are identified from the burning of cow dung compared to other fuels. These potential markers
predominantly contain nitrogen in chemical composition and overlap all potential markers for straw,
owing to the mixture of dried cow dung and crop residues in "cow dung cakes." Many nitrogen-
containing potential markers are found in straw and cow dung, such as $C_4H_5N$, $C_5H_5N$, $C_5H_7N$, and
$C_6H_7N$, which could likely be assigned to pyrrole, pyridine, methylpyrrole and methyl pyridines
respectively. Pyrolysis of the constituents in the crop residue is a probable pathway for these compounds
(Ma and Hays, 2008). Acetonitrile ($C_2H_3N$), acrylonitrile ($C_3H_3N$), propanenitrile ($C_3H_5NO$), and 4-
methylpentanenitrile ($C_6H_{11}N$) were found as potential markers for cow dung with generally higher FC
and higher relative contribution. Additionally, several series of nitrogen-containing homologues are
found, such as $C_{10}H_{11-21}NO$, $C_{12}H_{11-21}N$, $C_{11}H_{11-23}NO$ and $C_{15}H_{15-31}N$. These nitrogen-containing gases

have also been detected, especially in emissions from cow dung cake in India compared to fuelwood and are likely formed from the volatilization and decomposition of nitrogen-containing compounds within the cow dung cake. These compounds originate primarily from free amino acids but can also arise from pyrroline, pyridine, and chlorophyll (Stewart et al., 2021a).

Coal is also a distinct solid fuel compared to other biomass fuels in this study, showing a relatively lower correlation coefficient (Figure 2). Consequently, many series of $C_xH_y$-family homologues are found. Compounds with 9-12 carbon atoms, as shown in Figure 5 for coal burning, could be PAHs-related, such as $C_9H_8$ (1-Indene), $C_{10}H_8$ (naphthalene), $C_{10}H_{10}$ (1-methylnapthalene), $C_{12}H_{10}$ (acenaphthene), $C_{12}H_{12}$ (2,6-dimethylnaphthalene). The EFs of the potential markers also show an increasing trend with the decrease of MCE (Figure S13), which suggests EFs of the potential markers are not only dependent upon the type of fuel burnt but also on the burning condition.

## 4 Conclusions

In this study, we investigated emissions of organic vapors using Vocus during typical solid fuel combustion, including burning of beech logs, spruce/pine logs, spruce/pine branches and needles, straw, and cow dung and coal briquettes. Average EFs of CO, $CO_2$, organic vapors and PM were calculated. This work provides a comprehensive laboratory-based analysis of the chemical composition of organic vapors emitted from the different combustibles and different combustion phases. We discuss the prominent net combustion emissions from BB and identify new potential markers using the Mann-Whitney U test.

The results indicate that wood burning has higher organic vapors EFs compared to other fuels. The emissions varied significantly, ranging from 4.8 to 74.2 g kg$^{-1}$, depending on the combustion phases and solid fuel types. Despite the slight difference in modified combustion efficiency (MCE) for some experiments, the increasing EFs for organic vapors were observed with lower MCE. Moreover, the EFs of these SOA precursors are much higher than the primary biomass-burning organic aerosol (BBOA), which suggests a higher potential for SOA formation. Distinct particulate/gas emissions at different combustion phases are observed for stove burning of beech logs: initial compositions of flaming or smoldering plumes were dominated by BC or OA, respectively, with much higher organic vapor emission in smoldering. The relative contribution of different classes showed large similarities between the combustion phases in beech logs stove burning, relative to the large change in EFs observed. Therefore, the enhanced EFs under smoldering conditions means there is a greater potential for SOA formation when compared to flaming conditions.

The $C_xH_yO_z$-family is the most abundant group (63.1% to 88.6%) for all solid fuels, followed by $C_xH_y$ (9.3% to 33.7%). A larger contribution of nitrogen-containing species ($C_xH_yN$ and $C_xH_yO_zN$) is found in cow dung burning, while coal burning has a higher contribution from the $C_xH_y$ families. Moreover, the VOC class was found to be the most abundant (58.7% to 87%), followed by the IVOC class (12.6% to 39.3%). Primary semivolatile/intermediate-volatility organic compounds (S/IVOCs) have been proposed as important SOA precursors from BB. Li et al. (2024) found that IVOCs from residential wood burning (~ 13% of total organic vapors) can contribute ~70% of the formed SOA. Overall, these data will help update the IVOC emission inventory and improve the estimates of SOA production. Specifically, these results demonstrate that open burning (e.g., wildfire) emissions have enhanced IVOC

EFs, suggesting that the SOA potential from open-burning sources is larger in comparison to their wood stove counterparts.

However, each source generally emits a wide spectrum of organic vapors species, leading to considerable overlap between organic vapors species from different sources. Based on the Mann-Whitney U, we selected species that were unique in certain emissions as possible potential markers for specific solid fuels and the dominant compounds for all biomass fuels. Due to minor natural emissions from spruce in summer, monoterpene ($C_{10}H_{16}$) and its fragment could be potential markers for all softwoods (i.e., pine logs and spruce/pine branches and needles in this study) in winter. More products of the pyrolysis of coniferyl-type lignin and the cedar pine needle extract could be found in spruce/pine branches and needles open burning (e.g., $C_{10}H_{14}O_2$, $C_{11}H_{14}O_2$, $C_{10}H_{10}O_2$). The prominent product ($C_9H_{12}O$) resulting from the pyrolysis of beech lignin is identified as the potential markers for beech log stove burning. Many series of nitrogen-containing homologues and nitrogen-containing species (e.g., acetonitrile, acrylonitrile, propanenitrile, methylpentanenitrile) are identified (e.g., $C_{10}H_{11-21}NO$, $C_{12}H_{11-21}N$, $C_{11}H_{11-23}NO$ and $C_{15}H_{15-31}N$), particularly from open burning of cow dung. Coal is a unique solid fuel compared to biomass and more PAHs-related potential markers are identified from coal burning with 9-12 carbon. These potential markers provide important support for future global or regional chemistry transport modeling and source apportionment. Overall, our study provides a comprehensive understanding of the molecular composition and volatility of primary organic compounds, as well as new insights into the identification of potential markers from the burning of solid fuels.

# Tables and figures

**Table 1** Average EFs of CO, $CO_2$, organic vapors and PM as well as MCE for 6 types of Solid fuel type.

| Solid fuel type | Carbon content | MCE | Emission factors (g $kg^{-1}$ fuel) | | | |
|---|---|---|---|---|---|---|
| | | | CO | $CO_2$ | organic vapors | PM |
| beech logs stove (n=6) | 0.46 | 0.96 ± 0.03 | 38.9 ± 25.9 | 1409.4 ± 177.1 | 74.2 ± 42.9 | 2.5 ± 1.7 |
| spruce/pine logs stove (n=5) | 0.46 | 0.97 ± 0.01 | 28.5 ± 14.3 | 1511.7 ± 68.5 | 44.9 ± 17.5 | 1 ± 0.6 |
| spruce/pine branches and needles open (n=3) | 0.46 | 0.99 ± 0.001 | 2.8 ± 0.8 | 1579.2 ± 29.7 | 39.8 ± 11.4 | 0.9 ± 0.4 |
| straw open (n=4) | 0.45 | 0.97 ± 0.01 | 24.4 ± 6.6 | 1488.4 ± 87.2 | 42.6 ± 33.7 | 2.8 ± 0.7 |
| cow dung open (n=5) | 0.45 | 0.95 ± 0.03 | 53.9± 27.2 | 1541.8 ± 50.2 | 4.8 ± 0.98 | 1.2 ± 0.61 |
| coal stove (n=5) | 0.49 | 0.96 ± 0.01 | 40.6 ± 12.6 | 1680.2 ± 32.7 | 11.5 ± 2.6 | 0.9 ± 0.3 |

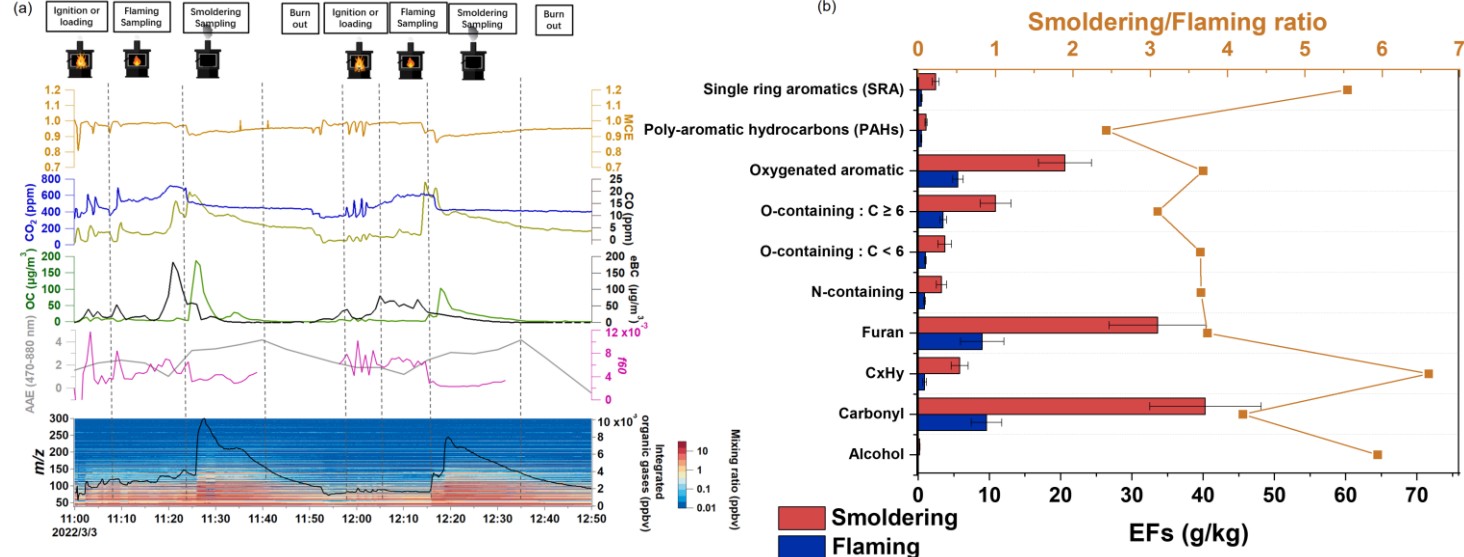

**Figure 1.** (a) Temporal profiles of mixing ratios measured by Vocus and evolution of CO, $CO_2$, AAE, $f60$, MCE and key aerosol compositions during burning cycles of beech logs stove burning (b) Geometric mean of the primary EFs for gas-phase species of different functional groups during flaming and smoldering phase, respectively (the flaming and smoldering was separated by the experimental record and calculated MCE). Error bars correspond to the sample geometric standard deviation of the replicates. The square represents the mixing ratio between smoldering and flaming. In this study, the MCE is used to indicate the flaming stage and smoldering and a significant decrease of MAC and $CO_2$ was observed from the flaming phase to the smoldering phase.


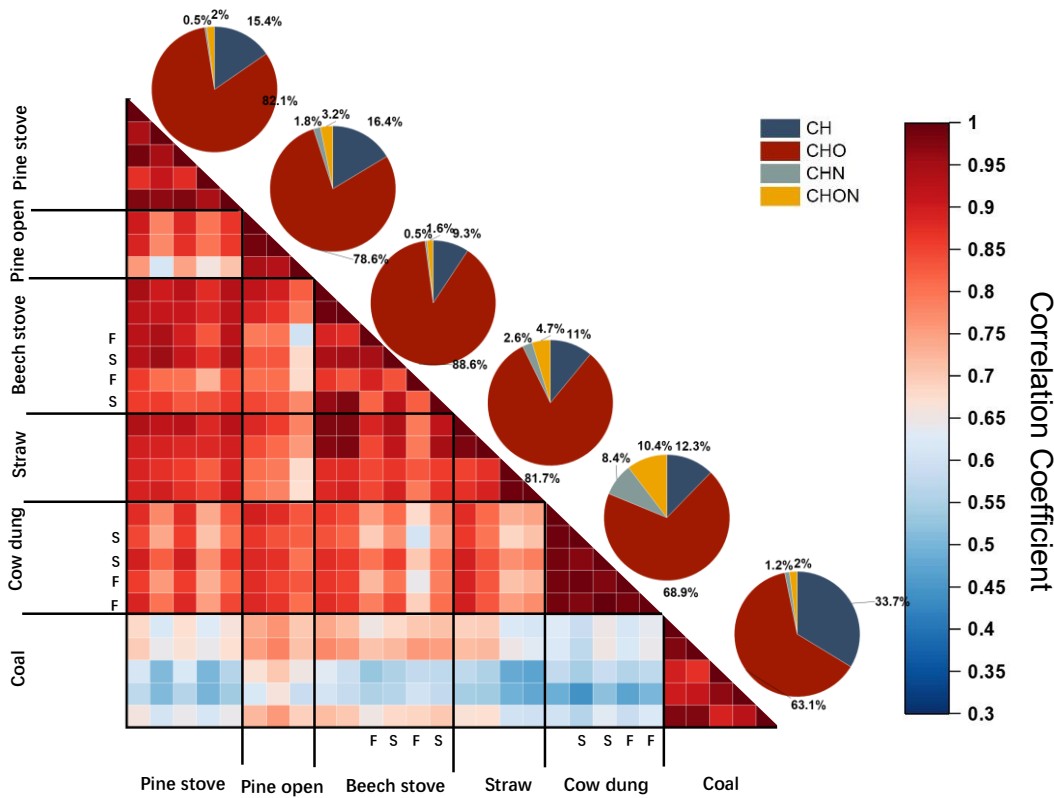

Figure 2. The correlation matrix of organic vapors measured with Vocus (F represents flaming phase and S represents smoldering phase and unmarked columns and rows represent mixtures of both flaming and smoldering phases). Pie charts showing the contribution of elemental families are on the diagonal.


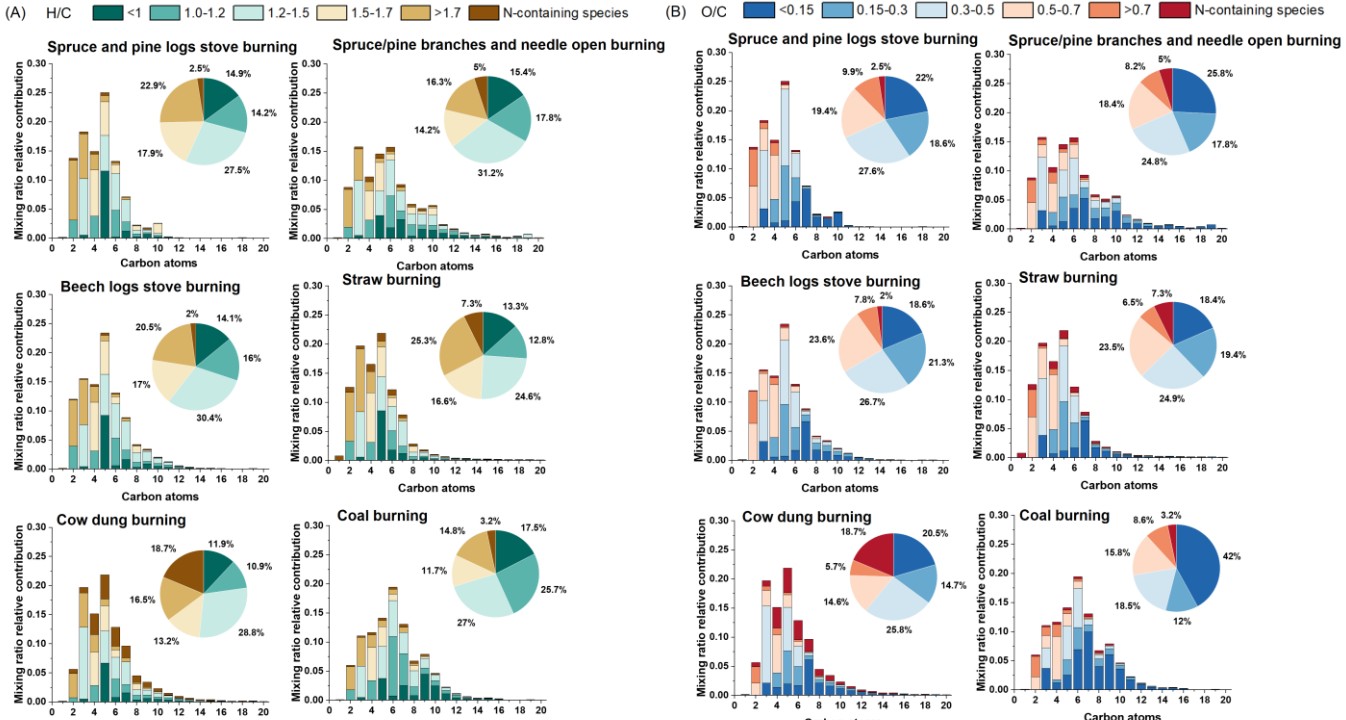

**Figure 3.** The average carbon distribution is colored by the H/C (**A**) and O/C (**B**) for non-N-containing species. The pie charts are the corresponding contribution of a range of H/C or O/C ratios.

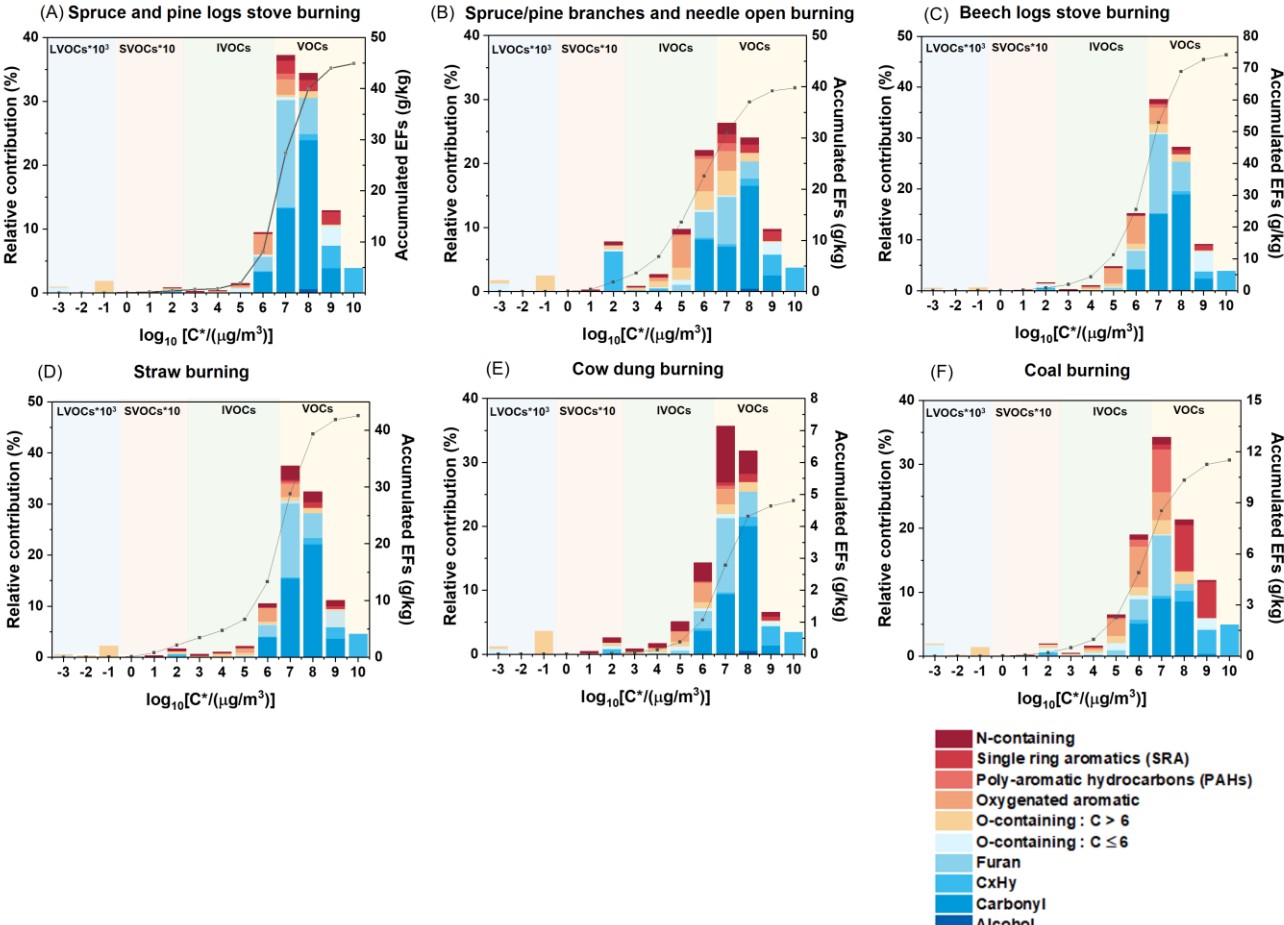

**Figure 4.** Volatility and average accumulated EFs (assume the average molecular weight of each bin are same)
the distribution of primary emissions as a function of binned saturation vapor concentration. Shaded areas indicate
the volatility ranges with units of μg m$^{-3}$: VOCs (yellow) as $\log_{10}(C^*) > 6.5$, IVOCs (blue) as $\log_{10}(C^*)$ between
6.5 to 2.5, semi-VOCs (SVOCs, green) as $\log_{10}(C^*)$ between 2.5 to - 0.5 and low-VOCs (LVOCs, orange) as
$\log_{10}(C^*) < - 0.5$. The relative contribution of LVOCs and SVOCs are multiplied by a factor of 1000 and 10,
respectively.

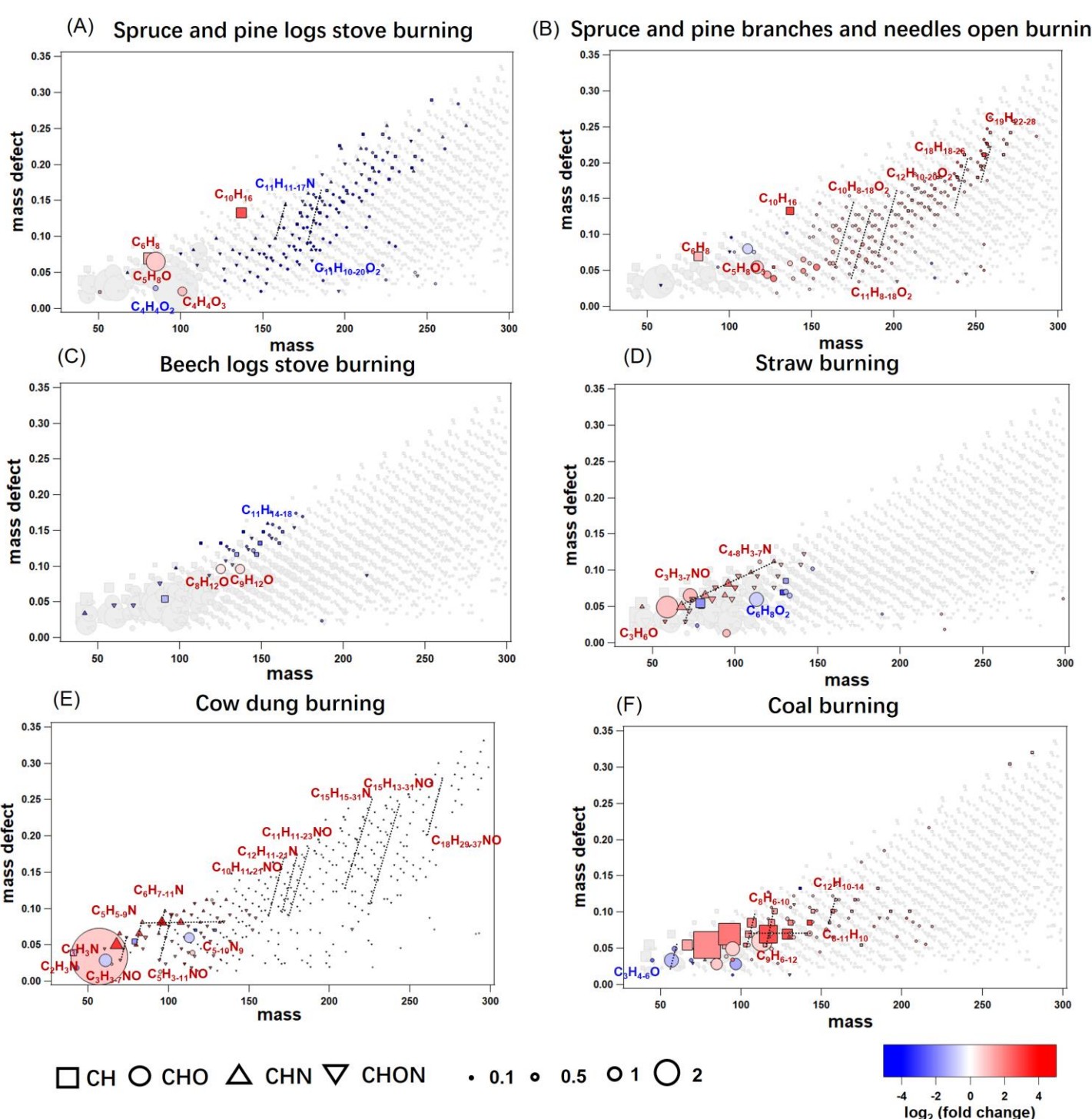

**Figure 5.** Mass defect plots identifying potential markers sized by the square root of fractional contribution (%) and colored by $\log_2$ (the fold change). The dashed line represents the series of homologues.

## Data availability

The data presented in the text and figures are available in the Zenodo online repository (https://doi.org/ 10.5281/zenodo.14204572).

## Author contributions

TTW, JZ, HL, KL, RKYC, EG, LK, DMB, and RLM conducted the burning experiments. TTW analyzed the data and wrote the paper. MB, ZCJD, LK, DMB, KL, RLM, IEH, HL, JGS, and ASHP participated in the interpretation of data.

## Competing interests

The authors declare that they have no conflict of interest.

## Acknowledgments

This work was supported by the Swiss National Science Foundation (SNSF) SNF grant MOLORG (200020_188624), an SNSF Joint Research Project (grant no. IZLCZ0_189883), the PSI career return fellowship, and the European Union's Horizon 2020 research, innovation programme under the Marie Skłodowska-Curie grant agreement No 884104 (PSI-FELLOW-III-3i) and ATMO-ACCESS. PSI's atmospheric simulation chamber is a facility of the ACTRIS ERIC and receives funding from the Swiss State Secretariat for Education, Research and Innovation (SERI grant).

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
