# Peer review of "Chemical characterization of organic vapors from wood, straw, cow dung, and coal burning"

_EGUsphere, 2024_

## Referee Comment (RC1)

Organic vapors from wood, straw, cow dung, and coal burning using Vocus PTR-TOF

Tiantian Wang et al.

**General comments**

This manuscript included interesting topic and monitored not only gaseous organic compounds but also other tracers such as $CO_2$, CO, and aerosols which can assess different six fuel types. The authors used several tools using monitoring data to identify characteristics of each fuel type. In the aspects of reduction and verification for their emissions on climate and air quality issue, this study would be valuable reference for research community in the future.

Even though its value, this manuscript was not well written.

First, this manuscript would be more suitable to 'measurement report' type rather than 'research article' (https://www.atmospheric-chemistry-and-physics.net/about/manuscript_types.html).

Next, the title of this manuscript doesn't cover and represent of whole manuscript. The title makes readers misunderstand that this paper focus on Vocus PTR-TOF instrument. However, the manuscript includes more than that. Therefore, it would be good to find appropriate title.

The terms were used too complicated to understand. There are several terms which seems like same or subtly different but used together without any clear categorizations (e.g. burning, burning type, fuels, fuel type, biomass burning, wood, and solid fuels).

Authors might assume many researchers can distinguish and realize those terms by themselves without clear explanation. However, in the scientific manuscript, clear definitions and unify the term in a whole manuscript are very important tools for making readers understand what authors would like to say.

Similarly, many of undefined (or unclear) terms were used in whole manuscript such as 'common compounds' and 'characteristic compounds' including 'selected characteristic compounds'. It was not clear what common compounds (and also characteristic compounds) mean and represent of.

For the results and discussion Sect., authors handled with lots of information and data all together. This absolutely contributes to research community as a good reference.

However, every section displayed very independently and did not link together without any explanation of purpose for each section. This might be very confusing for readers. Therefore, I would like authors to re-organize structures in the manuscript and make it appealing.

Unfortunately, the authors did not seem to give their attention meticulously. For example, some figures should be swamped (not be matched with each figure explanations), there are many abbreviations without full names, reference and explanation were not matched, etc.

Overall, even if a research topic and result are great, if authors cannot deliver the results efficiently, that papers would be difficult to be accepted. Because readers cannot read and fully understand the manuscript and its value.

My specific comments more focused on writing rather than paper's scientific logic and value. Also, there might be more terms/words and sentences I could not find out at the moment.

Hope authors consider specific comments and revise/improve its whole manuscript. And then re-submit the manuscript. After this, we can discuss it on a scientific level.

Regards,

**Specific comments**

1. P1 L19: Through whole manuscript, real-time gas-phase emissions analysis was not occupied large part. There is no timeseries data set without

2. P1 L25-27: Please explain explicitly with a clear sentence. Authors use this type of sentence in the whole manuscript. It would be good to make short sentences with clear sentence.

   - The CxHyOz family is the most abundant group (of what?),
   - but a greater contribution of nitrogen-containing species (than what?)

and CxHy families (related to polycyclic aromatic hydrocarbons) could be found in cow dung burning and coal burning, respectively.

This sentence is quite vague to read because authors use conjunctions couple of times such as 'but' and 'and' in one sentence.

3. P1 L28: Please explain explicitly with a clear sentence.
   - especially for spruce and pine branches and needles (39.3%), and coal (31.1%)
   - To: especially for spruce/pine branches and needles (39.3%) and coal (31.1%)

4. P1 L33: What does the characteristic product mean? It is only for $C_9H_{12}O$? Is it scientific term people use in this field? Is it different from 'characteristic compounds' on line 32? What is 'characteristic compounds'?

5. P2 L64: Author used the term of 'characteristic compounds'. But never define what it means. This is totally different from the term such as 'organic compounds'. Because readers can know what 'organic' is that it is not necessary to explain. However, 'characteristic' means a lot. Therefore if it is scientific term, authors should explain what it is or what kind of gases are belonging to this category. If it is not a scientific term, it is wrong expression.

6. P3 L93 and L97: Author mentioned that 'Vocus PTR time-of-flight mass spectrometer (hereafter Vocus)' but in L97, mentioned again Vocus PTR-TOF.

   If it is same word, please add TOF in L93 and swamp 'Vocus PTR-TOF' to 'Vocus'.

7. P3 L113: The title of section 'Fuel and burning set-up' can 'Fuel and burning types set-up'. Without 'types', the title sounds like physical set-up (of course, it includes the concept though.)

8. P3 L117: Author mentioned six fuel type but did not explain why those were chosen for experiment.

9. P3 L121 to 130: Through the whole manuscript, it is very difficult to find the linkage those explanations to the results. More efficient way to deliver how to categorize the burning type is to suggest 'Six burning type with different fuels' first. For example, *'with those six different fuels, we categorized six burning types for this experiment. 1) beech logs stove, 2) spruce and pine logs stove, 3) spruce and pine branches and needles open, 4) straw open, 5) cow dung open and 6) coal stove. Among the list above, 1) and 2) are representative of residential wood burning..... (Table 1). '*

10. P4 L125: There are lots of commas so that it makes readers difficult to understand. On this manuscript, there are many sentences with similar structures to this sentence.

    'Combustion of agricultural waste, straw, and a mixture of fresh spruce and pine branches and needles were in an open stainless-steel cylinder measuring 65 cm in diameter and 35 cm in height.'

    Authors can divide into two sentences or make it simple. Example below:
    For combustion of agricultural waste, ....spruce/pine branches and needles...'

11. P4 L132: Section title is not clear. It can be sampling and analysis method.
12. P4 L134: Pure Air with N2, O2, Ar? The purity percentage?
13. P4 L137: diluter ... Diluted by what?
14. P4 L151: Black Carbon (BC)?
15. P4 L152: A LI-7000 CO2 analyzer (LI-COR) provided continuous measurements of carbon dioxide (CO2).
    Is it same instrument of CO2/CO monitor in Figure S1? In this case, please add carbon monoxide as well.

16. P5 L160: The Vocus was calibrated before and after......
    For the reliable data for other species, it would be good to suggest calibration strategy to other instrument as well.

17. P5 L162 and L165: Please clarify pure air and zero air difference. If it is same, please unify the term. And also make it clear that only dried air used for background measurement.

18. P5 L173: Why did not authors make the title clear? In section 2.4, the title is very clear what will be discussed in the section. However for section 2.3, the title is very vague.

19. P5 L176: excess mixing ratio above what? If you use the excess mixing ratio of CO and CO2 above background air, please add information of background air (where those data you download or how you measure it.) And also this methods in Equation 1. seems like little different from the reference (Ward and Radke, 1993).

20. P5 L179: The sentence seems like not organized well.

If authors would like to add instrument information inside blank, please add all of instrument information. If authors would like to describe species, please unify the species all. And clarify that meaning of conventional gases.

21. P5 L182: mass concentration.

Author use the unit as combination of mixing ratio and mass concentration. However it is different unit.

(mass) mixing ratio: Number of the mass of the target gas (species) per mass of air (possible units are ppmm (also ppmw) = parts per million of air molecules by mass (weight), etc.). A specification whether it refers to dry or moist air is required.

However, mass concentration is mass/volume.

Please clarify all units.

22. P5 L182: $\Delta OC$: This was not discussed in section 2.2.
23. P5 L184: OM also was not discussed at all.
24. P6 L203: In this study, the mixing ratio relative contribution for more than 1500 species from six different fuels 204 for all 28 test burns was quantified by using Vocus.

It is difficult to understand the meaning of 'mixing ratio relative contribution'. Relative contribution to what?

25. P6 L204: characteristic compounds. This term never mentioned before in this manuscript.
26. P6 L213-L218: make the sentences clear.

*However, due to the similarity in fuel types between burning spruce and pine logs, as well as spruce and pine branches and needles, they were categorized as separate fuel sources for this test and not compared with each other but were only compared with the other four types of fuels.*

- It would be good to make it simple: for this test, 'spruce and pine logs' and 'spruce/pine branches and needles' were in the same fuel type category due to similar characteristics.
- And then readers can have questions, if so (they have same characteristics), why did authors use all of them as different fuels at the beginning?
- If author define 6 different fuel type, please keep the term of 'fuel type' in the whole manuscript. Please don't change to fuel sources or other. It makes confusions. Using term should be clear and same. For burning type as well.
- The sentence below is difficult as well.

*Similarly, due to the composition of cow dung "cakes," which comprise a mixture of dried cow dung and crop residues and a relatively high correlation between cow dung and straw (Figure 1), the Mann-Whitney U test was carried out without accounting for the presence of the other fuels.*

27. P7 L226: Emission factors from solid-fuel combustion to 'The characteristics of EF and MCE from different fuel types.' ? make it more clear.
28. P7 L237: did author mention that NMOG is primary organic gases in the above manuscript? It would make easier to readers if you mentioned that NMOG is primary organic gases in the introduction. or Just using NMOG is necessary in whole manuscript. It is very important that using same term in whole manuscript for readers.
29. P7 L243: The slightly higher EFs for pine and spruce wood burning can be attributed to the more extensive analysis of NMOGs in our study compared to previous research (37.3 245 g/kg) (Hatch et al., 2017).

- Can we say this value is higher than pervious study even though the difference is in the uncertainty levels?

30. P7 L249: emissions. Does it mean EF? Authors know the difference between emissions and EF.

31. P7 L250: emission factors. If authors started using EF in place of emission factors, please use EF.
32. P7 L253: title again. Authors never mentioned about primary organic vapors in previous sections. It can be 'The characteristics of NMOG chemical composition from different fuel types'.
33. P7 L254: I don't think section 3.2.1 can be overview. In this section, general characteristics can be handled.
34. P7 L255: combustion fuel source mean burning type? Please keep same term.
35. P7 L257: what is primary NMOGs? So is it different from just NMOGs?
36. P7 L261: biomass samples. What kind of burning type are belonging to biomass samples. Please defined clearly.
37. P8 L273: What is H/C ratio?
38. P8 L274: What is O/C ratios?
39. P8 L278: Suggested its value. Science paper should suggest certain number otherwise 'slightly lower' means very subjective.
40. P8 L290-L301: I suggested this paragraph can come up the beginning of this section. And then authors can easily explain Figure 1 and Figure 2. On the other hand, I also wondered NMOG will be categorized only primary organic gases and secondary as well.
41. P9 L303: Hard to understand this sentence.
    Based on the log10C* values of all organic compounds parameterized with the modified approach of Li et al. (2016) described in Sect. 2.4.
42. P9 L308: What kind of burning type are included in biomass burning emissions?
43. P9 L311: For all burns. Did author mean for all experiment or for all burning types?
44. P9 L320-323: I think the reference cannot support author's finding.
    In general, spruce and pine branches and needles open burning released a higher proportion of IVOCs (39.3%) into the gas phase compared to stove logs burning (12.6% and 322 23.9%). Pallozzi et al. (2018) also reported a similar result, showing that needle/leaf combustion released a greater amount of volatile organic compounds into the atmosphere than branch combustion.

45. P9 L332: VBS. What does VBS mean?
46. P9 L333: wood burning. Did author define wood burning and biomass burning somewhere? What kind of burning type are in this category? And why do authors want to compare two stages? why do authors only analyze wood burning for this comparison?
47. P9 L335-337: I don't agree with authors point of view. CO increased when the stage was changed from flaming to smouldering.
    *In the top panel, the MCE is used to indicate 336 the flaming stage with a significant CO2 enhancement, while the smoldering stage exhibits high levels of CO.*

48. P9 L338: What AAE means?
49. P9 L339: *f*60. Have ever author described what it is in the manuscript or table?
50. P9 L341: BB. What does it mean?
51. P10 L350: (31.4 g/kg) and (121.9 g/kg). Please add standard deviation or uncertainty ± value.
52. P10 L351: Large similarity. I cannot understand large similarities. Did authors mean similar trend?
53. P10 L352: Figure S7. doesn't have information that what each of #c and #o represent for and also its explanation does not match to the Figure.
54. P10 L356: What does OVOC stand for?
55. P10 L356: Figure S8. I assumed Figure S8 was not the figure Author would like to show.
56. P10 L365: What does 'common' mean?
57. P10 L382: Unify the term. there is no 'wood' in Figure 1.
58. P11 L395: solid-fuel combustion. This is very confusing that solid fuel, biomass burning, fuel type, wood... all things are very tangled so that hard to understand. Please keep same term and define each term clearly.
59. P11 L398: add number of table.
60. P11 L399: Very confusing between characteristic compounds and selected characteristic compounds. What are the differences between two of them?
61. P11 L401: In contrast, compounds from open burning of straw and cow dung
    Does it different form characteristic compounds and selected characteristic compounds?

62. P12 L448: l solid fuel combustion, including residential burning (beech logs, a mixture of spruce and pine logs, and coal briquettes) and open combustion (spruce and pine branches and needles, straw, and cow dung).
    - Author also investigated biomass burning and woods? did you use biomass burning same to solid fuel?
    - The burning type category is very important here. Please use it clearly.
63. P13 L481: Still have questions of characteristic compounds and the common compounds.
64. Figures and table.
    - Figures label fonts were too small to read.
    - Figure labels and explanations in the captions should be same to the explanation in the manuscript.
    - Figure 4. What is key aerosols? Figure 4(b) the colour bar of Smouldering and Flaming can be moved to right side of the panel.
65. Reference.
    - Please give the space between the references at least.

---

## Author Comment (AC1)

**Reviewer #1**

I read this paper with great interest and congratulate the authors on an impressive set of results. However, the value of this study for the scientific community could be improved immensely if the authors would add in the supplement a Table with emission factors at least for the 100 most important VOCs as well as aerosol components like BC and OC. The most useful format of this supplement would be an Excel spreadsheet.

We would like to thank the reviewer for the comments and suggestions to improve the current work. We will have the reviewer comments in black, address the comments in blue, and modified sentences in red.

We have attached another table with more details of the emission factors for all organic vapors. Besides, we also added the BC and OC emission factor for the experiments we measured.

1. One of the most surprising results is the very clean combustion of the dung cakes. This is in strong contrast to all previous studies. The nine studies in my database give an average MCE of 0.88 +- 0.04 (Andreae, 2019), while this study gives 0.98 +- 0.01. It would be interesting to see a discussion of what may explain this difference.

**Response:** We apologize for the error in the emission factor during the cow dung experiments. We have corrected the calculations. However, the average MCE (ranging from 0.89 to 0.97, with an average value of $0.95 \pm 0.03$) is still higher than the values reported in your study ($0.88 \pm 0.04$). In our study, the average MCE was calculated based on the real-time emissions from cow dung burning. Due to the lower concentration of organic vapors produced during cow dung combustion, we only selected data from periods with higher concentrations, which allowed us to detect more organic compounds for subsequent marker-selected analysis. This selection likely contributed to the relatively higher average MCE compared to your study. Additionally, I found other studies reporting higher MCE values. For example, Pervez et al. (2019) measured the MCE of dung cake burning in India, which ranged from 0.91 to 0.99. They found that MCE values from cow dung burning could range from smoldering-dominated combustion (MCE = 0.73) to flaming-dominated combustion (MCE = 0.99). The lower MCE during smoldering was achieved at a lower furnace temperature, while the higher MCE during flaming was reached at a temperature of 800°C.

2. On a minor note, the reference to the now outdated Andreae & Merlet (2001) should be replaced by the updated paper:

Andreae, M. O., Emission of trace gases and aerosols from biomass burning – an updated assessment: Atmos.

Chem. Phys., 19, 8523-8546, doi:10.5194/acp-19-8523-2019, 2019.

**Response:** We have updated the reference in the manuscript.

**Reference:**

Pervez, S., Verma, M., Tiwari, S., Chakrabarty, R. K., Watson, J. G., Chow, J. C., Panicker, A. S., Deb, M. K., Siddiqui, M. N., and Pervez, Y. F.: Household solid fuel burning emission characterization and activity levels in India, Science of The Total Environment, 654, 493-504, 2019.

---

## Author Comment (AC2)

**Reviewer #2**

**General comments**

This manuscript included interesting topic and monitored not only gaseous organic compounds but also other tracers such as CO2, CO, and aerosols which can assess different six fuel types. The authors used several tools using monitoring data to identify characteristics of each fuel type. In the aspects of reduction and verification for their emissions on climate and air quality issue, this study would be valuable reference for research community in the future. Even though its value, this manuscript was not well written. First, this manuscript would be more suitable to 'measurement report' type rather than 'research article' (https://www.atmospheric-chemistry-andphysics.net/about/manuscript_types.html).

**Response:** We would like to thank the reviewer for the comments and suggestions to improve the current work. We will have the reviewer comments in black, address the comments in blue, and modified sentences in red.

We have restructured the manuscript. Our measurements not only provide key parameters of combustion (e.g., emission factors) but also offer the ability to identify and quantify rarely measured and previously unidentified organic vapor emissions, particularly those in the chemically complex low-volatility fraction. These insights are crucial for advancing the current understanding of the impact of solid fuel combustion on air quality and climate. Moreover, we applied the Mann-Whitney U test to biofuels and coal, allowing us to identify specific potential new markers for these fuels based on Vocus measurements. These markers serve as valuable references for field campaign studies focused on source apportionment.

Given the depth and significance of our findings, we believe that this work is more appropriate as a "research article" rather than a "measurement report". It makes a substantial contribution to the scientific community's understanding of emissions and their implications.

Next, the title of this manuscript doesn't cover and represent of whole manuscript. The title makes readers misunderstand that this paper focus on Vocus PTR-TOF instrument. However, the manuscript includes more than that. Therefore, it would be good to find appropriate title.

**Response:** The main instruments we used for organic vapor measurement in this study is Vocus-PTR. However, the study also included the data from other instruments (e.g., $CO_2$, BC) for emission factor calculation and interpretation of different combustion phase. Thus, we deleted "Vocus PTR-TOF" in the tile and change it to "Chemical characterization of organic vapors from wood, straw, cow dung, and coal burning".

The terms were used too complicated to understand. There are several terms which seems like same or subtly different but used together without any clear categorizations (e.g., burning, burning type, fuels, fuel type, biomass burning, wood, and solid fuels). Authors might assume many researchers can distinguish and realize

those terms by themselves without clear explanation. However, in the scientific manuscript, clear definitions and unified term through a whole manuscript are very important tools for making readers understand what authors would like to say. Similarly, many of undefined (or unclear) terms were used in whole manuscript such as 'common compounds' and 'characteristic compounds' including 'selected characteristic compounds'. It was not clear what common compounds (and also characteristic compounds) mean and represent of.

**Response:** We agree that clear definitions and consistent usage of terms are essential for ensuring that our readers can easily understand the distinctions we are making.

1) Terminology Consistency: We have revised the manuscript to ensure that the term "solid fuel type" is used consistently throughout. To prevent any confusion, we have avoided using terms like "fuel sources" or any other variations interchangeably.

2) Clarification of Solid Fuels: In this study, solid fuels are defined to include both biofuels (such as beech logs, spruce/pine logs, spruce/pine branches and needles, straw, cow dung) and coal. We have added this classification to Section 2.1 to ensure that readers clearly understand what we mean by solid fuels in the context of this research.

3) Removal of Ambiguous Terms: To avoid ambiguity, we have removed all instances of the term "characteristic compounds" from the manuscript. Instead, we now refer to the selected substances as "potential markers" based on the statistical methods used, similar to those employed by Zhang et al. (2023). This change helps in providing a more precise scientific explanation.

4) Clarification of "Common" Markers: We have clarified that the term "common" refers to potential markers that are applicable across all biomass fuels, rather than a specific type of biomass fuel. The Mann-Whitney U test was performed to identify potential markers among different types of solid fuels. However, in this study, biomass fuels (such as logs, branches, needles, straw, and cow dung) were analyzed separately from coal due to their distinct characteristics. To address this distinction, we characterized the dominant compounds across various biomass fuels by setting a threshold (relative mixing ratio contribution ≥ 0.1%) for compounds that are not potential markers of one specific biomass fuels. This approach allowed us to identify compounds that are more readily detectable in complex environments. Recognizing that "common" is not a precise scientific term, we have removed it from the manuscript entirely to avoid any misunderstanding.

These revisions should help ensure that the terminology used in our manuscript is both clear and scientifically accurate, allowing readers to fully grasp the distinctions we are making in our study.

For the results and discussion Sect., authors handled with lots of information and data all together. This absolutely contributes to research community as a good reference. However, every section displayed very

independently and did not link together without any explanation of purpose for each section. This might be very confusing for readers. Therefore, I would like authors to re-organize structures in the manuscript and make it appealing.

**Response:** We have restructured the manuscript. We first discuss about "The characteristics of EF and MCE from combustion" and we found the average emission factors for organic vapors from different solid fuels have significant difference, depending on the combustion phases and solid fuel types. And then based on above results, we deeper analysis the relationship between MCE and combustion phase (smoldering and flaming) and the chemical composition of organic vapors from different solid fuels (Section 3.2 and Section 3.3). We observed that the chemical composition between different solid fuels is significant (O/C, H/C and relative contribution of different functional group, volatility distribution). Thus, in the Section 4, by using Mann-Whitney U test to biofuels and coal, we would like to find the dominant compounds for all biofuels and identify specific potential new markers for solid fuels based on Vocus measurements.

Thank you for your valuable feedback regarding the organization of the Results and Discussion sections. We understand that presenting a large amount of data and information can be overwhelming for readers, and it's crucial that each section is cohesively linked to ensure clarity and purpose. We have reorganized the manuscript to create a more logical flow and to clearly connect each section. We now begin with a discussion on "The characteristics of EF and MCE from combustion" (Section 3.1). In this section, we highlight the significant differences in average emission factors for organic vapors from various solid fuels, which depend on both combustion phases and fuel types. Building on these findings, we delve deeper into the relationship between EFs, combustion phase (smoldering and flaming), and the chemical composition of organic vapors from different solid fuels in Sections 3.2 and 3.3. Here, we show that the chemical compositions (O/C, H/C ratios, relative contributions of different functional groups, and volatility distribution) vary significantly among the different solid fuels. Finally, in Section 4, we apply the Mann-Whitney U test for all solid fuel types. This analysis aims to identify specific potential new markers for these fuels based on Vocus measurements. This section builds directly on the findings from the earlier sections and serves to tie the entire discussion together.

These revisions ensure that each section of the manuscript is interconnected, making the study more coherent and accessible. We believe this restructuring enhances the manuscript's appeal and clarity, allowing readers to better understand the progression of our analysis and the conclusions drawn.

Unfortunately, the authors did not seem to give their attention meticulously. For example, some figures should be swapped (not be matched with each figure explanations), there are many abbreviations without full names, reference and explanation were not matched, etc.

**Response:** Thank you for your meticulous review and for pointing out these critical issues. We have addressed

each concern: Figures in Supplement: We have thoroughly reviewed and corrected the placement of figures in the manuscript. The figures are now correctly matched with their corresponding explanations, ensuring that each visual element accurately supports the text.

Abbreviations: We have systematically checked for all abbreviations used throughout the manuscript. We have introduced the full names upon first use.

References and Explanations: We have carefully reviewed all references and their corresponding explanations. Any mismatches have been corrected to ensure that each reference accurately supports the related content.

**Specific comments**

1.  P1 L19: Through whole manuscript, real-time gas-phase emissions analysis was not occupied large part. There is no timeseries data set without flaming/smouldering stage explanation.

**Response:** We conducted real-time combustion experiments for all solid fuels, and as part of our analysis, we focused on the spruce/pine logs-burning experiment to illustrate the time series data between the flaming and smoldering phases. This example was chosen because the emission factors and other particulate data, such as black carbon (BC) and organic carbon (OC), show significant differences between these two phases. In response to your feedback, we have removed the term "real-time" from the abstract to reflect the content more accurately. However, we have retained the mention of "real-time" in the methods section to clearly describe the experimental setup and procedures.

2. P1 L25-27: Please explain explicitly with a clear sentence. Authors use this type of sentence in the whole manuscript. It would be good to make short sentences with clear sentence.

- The CxHyOz family is the most abundant group (of what?),

- but a greater contribution of nitrogen-containing species (than what?)

and CxHy families (related to polycyclic aromatic hydrocarbons) could be found in cow dung burning and coal burning, respectively. This sentence is quite vague to read because authors use conjunctions couple of times such as 'but' and 'and' in one sentence.

**Response:** We have rephrased this sentence and separated it into two sentences (Line 24-29).

The $C_xH_yO_z$ family is the most abundant group of the organic vapor emitted from all SF combustion. However, among these SF combustions, a greater contribution of nitrogen-containing species and $C_xH_y$ families (related to polycyclic aromatic hydrocarbons) is observed in the organic vapors from cow dung burning and coal burning, respectively.

3. P1 L28: Please explain explicitly with a clear sentence.

- especially for spruce and pine branches and needles (39.3%), and coal (31.1%)

- To: especially for spruce/pine branches and needles (39.3%) and coal (31.1%)

**Response:** We have rephrased this sentence and separated it into two sentences (line 30-32).

Intermediate volatility organic compounds (IVOCs) constituted a significant fraction of emissions in solid fuel combustion, ranging from 12.6% to 39.3%. This was particularly notable in the combustion of spruce/pine branches and needles (39.3%) and coal (31.1%).

4. P1 L33: What does the characteristic product mean? It is only for C9H12O? Is it scientific term people use in this field? Is it different from 'characteristic compounds' on line 32? What is 'characteristic compounds'?

5. P2 L64: Author used the term of 'characteristic compounds'. But never define what it means. This is totally different from the term such as 'organic compounds'. Because readers can know what 'organic' is, it is not necessary to explain. However, 'characteristic' means a lot. Therefore if it is scientific term, authors should explain what it is or what kind of gases are belonging to this category. If it is not a scientific term, it is wrong expression.

**Response:** To avoid ambiguous expressions caused by non-scientific terms, we have removed all instances of the term "characteristic compounds" from the manuscript. Instead, we have used the same statistical methods as Zhang et al. (2023) to select substances that are statistical outliers relative to other emission sources and, following this definition, we refer to the selected substances that are statistical outliers as "potential markers." Due to the similarities among different types of wood burning (open and stove burning), and in comparison to spruce burning, we identified only one new potential marker compound, $C_9H_{12}O$, which originates from the pyrolysis of beech lignin. However, other compounds resulting from the pyrolysis of coniferyl-type lignin could also be considered potential markers for all types of wood burning.

6. P3 L93 and L97: Author mentioned that 'Vocus PTR time-of-flight mass spectrometer (hereafter Vocus)' but in L97, mentioned again Vocus PTR-TOF. If it is same word, please add TOF in L93 and change from 'Vocus PTR-TOF' to 'Vocus'.

**Response:** We have revised the manuscript for consistency. In line with your suggestion, we have added "TOF" to the description to read "Vocus PTR-TOF mass spectrometer (hereafter Vocus)." We have also changed "Vocus PTR-TOF" to "Vocus" to ensure uniformity throughout the text.

7. P3 L113: The title of section 'Fuel and burning set-up' can 'Fuel and burning types set-up'. Without 'types', the title sounds like physical set-up (of course, it includes the concept though.)

**Response:** We have updated the section title to "Fuel and burning types" to better reflect the content and avoid any ambiguity regarding the physical set-up.

8. P3 L117: Author mentioned six fuel type but did not explain why those were chosen for experiment.

9. P3 L121 to 130: Through the whole manuscript, it is very difficult to find the linkage of those explanations to the results. More efficient way to deliver how to categorize the burning type is to suggest 'Six burning type with different fuels' first. It would be good if authors suggest simple table for it. For example, 'with those six different fuels, we categorized six burning types for this experiment. 1) beech logs stove, 2) spruce and pine logs stove, 3) spruce and pine branches and needles open, 4) straw open, 5) cow dung open and 6) coal stove. Among the list above, 1) and 2) are representative of residential wood burning….. (Table 1). '

10. P4 L125: There are lots of commas so that it makes readers difficult to understand. On this manuscript, there are many sentences with similar structures to this sentence. 'Combustion of agricultural waste, straw, and a mixture of fresh spruce and pine branches and needles were in an open stainless-steel cylinder measuring 65 cm in diameter and 35 cm in height.' Authors can divide into two sentences or make it simple.

**Response:** We have restructured this part and revised and separated the sentence to improve clarity by simplifying its structure.

The revised sentence now reads (Line 113-138):

The experiments were conducted at the Paul Scherrer Institute (PSI) in Villigen, Switzerland. The burning facility is part of the PSI Atmospheric Chemistry Simulation chambers (PACS). Real-time characterization of the primary gas and particle phase emissions was carried out during 28 test burns. Six solid fuels were studied (coal briquettes and biomass fuels: beech logs, spruce/pine logs, fresh spruce/pine branches and needles, dry straw, cow dung) with three to six replicate burns. Material in the beech, spruce, and pine fuels (e.g., logs and needles) was sourced from a local forestry company in Würenlingen, Switzerland. Cow dung cakes (a mixture of cow dung and straw) were collected from Goyla Dairy in Delhi, India. Coal briquettes were purchased from Gansu, China (Ni et al., 2021; Klein et al., 2018).

With those six different fuels, we categorized six burning types for this experiment. 1) beech logs stove, 2) spruce/pine logs stove, 3) spruce/pine branches and needles open, 4) dry straw open, 5) cow dung open and 6) coal stove. We selected these six solid fuels and conducted emissions tests to simulate certain types of burning found in the atmosphere. Among the list above, 1) beech logs stove and 2) spruce/pine logs stove are representative of residential wood burning, which are burned separately in a stove, consistent with the materials used in two previous articles (Bertrand et al., 2017; Bhattu et al., 2019). To represent forest fires or wildfire and agricultural field combustion, 3) a mixture of fresh spruce/pine branches and needles and 4) straw were combusted in an open stainless-steel cylinder (65 cm in diameter and 35 cm in height). Traditional cooking and heating practices in regions like India are represented by 5) cow dung cakes open burning by using half-open stoves (Loebel Roson et al., 2021). Finally, traditional cooking and heating practices in rural regions of developing countries are represented by 6) coal stove burning in a portable cast iron stove purchased from the local market (Liu et al., 2017). Of course, these conditions do not fully accurately represent the conditions found in actual fires, which consistent of a variety of burning species (e.g., trees, underbrush, peat soils, etc.…), but represent laboratory burning conditions.

11. P4 L132: Section title is not clear. It can be sampling and analysis method.

**Response:** In Section 2.2, we introduce the experimental design for the burning experiment and describe the

instruments used in this study. In Section 2.3, we present the analysis methods. We suggest that the section title could be "Experimental setup and instrumentation".

12. P4 L134: Pure Air with N2, O2, Ar? The purity percentage?

**Response:** We have added a sentence to describe the pure air generator and provided two references that used this generator in chamber experiments (Line 141-145).

The zero air was provided by a zero air generator (737-250 series, AADCO Instruments, Inc., USA) for cleaning and dilution (Heringa et al., 2011; Bruns et al., 2015). The zero air generator takes ambient air and scrubs particulates and volatile organic compounds from the air leaving a mixture that is largely made up of $N_2$, $O_2$, and Ar at ambient concentrations. Other trace gases are scrubbed to lower than atmospheric concentrations including $CO_2$ (< 80 ppb) and $CH_4$ (< 40 ppb).

13. P4 L137: diluter … Diluted by what?

**Response:** We have rephrased this sentence to clarify (Line 147-150).

Once a burn is initiated from the various combustibles, emissions are sampled from the chimney through a heated line (473 K). The emissions (both gas and particle phases) are then diluted by two Dekati diluters (DI-1000, Dekati Ltd.) which dilutes the emissions by a factor of ∼ 100 (473 K, DI-1000, Dekati Ltd.).

14. P4 L151: Black Carbon (BC)?

**Response:** We added the "BC".

15. P4 L152: A LI-7000 CO2 analyzer (LI-COR) provided continuous measurements of carbon dioxide (CO2). Is it same instrument of CO2/CO monitor in Figure S1? In this case, please add carbon monoxide as well.

**Response:** We added the instrument information for CO analyzer (Line 171-173).

A LI-7000 $CO_2$ analyzer (LI-COR) and APMA-370 CO analyzer (Horiba) provided continuous measurements of carbon dioxide ($CO_2$) and carbon monoxide (CO), respectively.

16. P5 L160: The Vocus was calibrated before and after…… For the reliable data for other species, it would be good to suggest calibration strategy to other instrument as well.

**Response:** The Vocus is the primary instrument used in this study, and we have not included extensive details about its calibration for the chamber experiments.

However, other instruments (e.g., AMS, AE, SMPS) are widely used in atmospheric studies. Therefore, we have not provided detailed descriptions of the calibration processes for these additional instruments. Instead, we have briefly added introduction of the maintenance and calibration in the section 2.2 and cited more references related to the calibration methods applied to these instruments (Line 157-175).

Numerous instruments were connected after the second dekati diluter for the characterization of both the particulate and gaseous phases. A Scanning Mobility Particle Sizer (SMPS, CPC 3022, TSI, and custom-built DMA) provided particle number size distribution information and calibrated by using polystyrene latex (PSL) particle size standards (Wiedensohler et al., 2018; Sarangi et al., 2017). The non-refractory particle composition was monitored by a high-resolution time-of-flight aerosol mass spectrometer (HR-ToF-AMS, Aerodyne Research Inc.). AMS data were processed using SQUIRREL (SeQUential Igor data RetRiEvaL v. 1.63; D. Sueper, University of Colorado, Boulder, CO, USA) and PIKA (Peak Integration and Key Analysis v. 1.23) to obtain mass spectra of identified ions in the $m/z$ range of 12 to 120. OC (organic carbon) is derived from the ratio of organic mass (OM) to OC (OM/OC) determined with high-resolution AMS analysis (Canagaratna et al., 2015). In the AMS mass spectra, the fraction of $m/z$ 60 ($f$60) represents the ratio of levoglucosan-like species (Schneider et al., 2006; Alfarra et al., 2007). AMS was calibrated for ionization efficiency (IE) by a mass-based method using $NH_4NO_3$ particles(Tong et al., 2021). Black carbon (BC) was measured with an Aethalometer (Magee Scientific Aethalometer model AE33) (Drinovec et al., 2015) with a time resolution of 1 minute. The maintenance and calibration are given in the AE33 user manual – version 1.57. A LI-7000 $CO_2$ analyzer (LI-COR) and APMA-370 CO analyzer (Horiba) provided continuous measurements of carbon dioxide ($CO_2$) and carbon monoxide (CO), respectively. The concentrations of total hydrocarbons (THC) and methane ($CH_4$) were monitored using a flame ionization detector monitor (THC monitor Horiba APHA-370).

17. P5 L162 and L165: Please clarify pure air and zero air difference. If it is same, please unify the term. And also make it clear that only dried air used for background measurement.

**Response:** We have reviewed the terms and confirm that "pure air" and "zero air" refer to the same concept in this context. We have unified the terminology to use "zero air" throughout the document. Additionally, we have clarified that only dried zero air was used for Vocus background measurements.

18. P5 L173: In section 2.4, the title is very clear what will be discussed in the section. However, for section 2.3, the title is very vague.

**Response:** We have combined Sections 2.3 and 2.4, as both sections discuss the calculation of key parameters: MCE, EFs, and volatility. This merger provides a clearer and more cohesive structure for the content.

19. P5 L176: excess mixing ratio above what? If authors use the excess mixing ratio of CO and CO2 above background air, please add information of background air (where those data download or how measure it.) And also this methods in Equation 1. seems like little different from the reference (Ward and Radke, 1993).

**Response:** We have corrected the reference and added the information of the background air (Line 197-198). Where $\Delta CO$, $\Delta CO_2$ are the mixing ratios of CO or $CO_2$ in excess of background (measured before the combustion), respectively (Christian et al., 2003).

20. P5 L179: The sentence seems very complicated. If authors would like to add instrument information inside

bracket, please add all of instrument information. If authors would like to describe species, please unify the species all. And clarify that meaning of conventional gases.

**Response:** We have rephrased this sentence to improve clarity and make it easier to understand (Line 201-204). The emission factors (EFs, g kg$^{-1}$) of species $i$ was calculated, following a carbon-mass balance approach (Andreae, 2019; Boubel et al., 1969; Nelson, 1982):

$$EF_i = \frac{m_i}{\Delta mCO + \Delta mCO_2 + \Delta mCH_4 + \Delta mNMOGs + \Delta mOC + \Delta mBC} \times \cdot W_C$$

Here $m_i$ refers to the mass concentration of species $i$. $\Delta mCO$, $\Delta mCO_2$, $\Delta mCH_4$, $\Delta mNMOGs$, $\Delta mOC$, and $\Delta mBC$ are the background-corrected carbon mass concentrations of carbon-containing species in the flue gas.

21. P5 L182: mass concentration.

Author used units as combination of mixing ratio and mass concentration. However, it is different unit. (mass) mixing ratio: Number of the mass of the target gas (species) per mass of air (possible units are ppmm (also ppmw) = parts per million of air molecules by mass (weight), etc.). A specification whether it refers to dry or moist air is required. mass concentration is mass/volume. Please clarify all units.

**Response:** The data used for MCE calculation are the CO and $CO_2$ mixing ratios. However, the emission factor calculation uses mass concentration data. The misunderstanding arose from using the same symbol for these two equations. We have corrected the symbol in the emission factor equation to clarify this distinction (Line 201-204).

The emission factors (EFs, g kg$^{-1}$) of species $i$ was calculated, following a carbon-mass balance approach (Andreae, 2019; Boubel et al., 1969; Nelson, 1982):

$$EF_i = \frac{m_i}{\Delta mCO + \Delta mCO_2 + \Delta mCH_4 + \Delta mNMOGs + \Delta mOC + \Delta mBC} \times \cdot W_C$$

22. P5 L182: ΔOC: This was not discussed in section 2.2.

23. P5 L184: OM also was not discussed at all.

**Response:** We have added the discussion and definition of OC and OM in section 2.2 (Line 165-166).

OC (organic carbon) is derived from the ratio of organic mass (OM) to OC (OM/OC) determined with high-resolution AMS analysis (Canagaratna et al., 2015).

24. P6 L203: In this study, the mixing ratio relative contribution for more than 1500 species from six different fuels for all 28 test burns was quantified by using Vocus. It is difficult to understand the meaning of 'mixing ratio relative contribution'. Relative contribution to what?

**Response:** The relative contribution of species $i$ was calculated as the mixing ratio of species $i$ divided by the mixing ratio of total organic vapor, multiplied by 100%. We have rephrased the sentence (Line 221-222).

In this study, the relative contribution of the mixing ratio for over 1,500 species from six different fuels was quantified across all 28 test burns using the Vocus.

25. P6 L204: characteristic compounds. This term never mentioned above in this manuscript.

**Response:** We have changed the term "characteristic compounds" to "potential markers" throughout the manuscript to ensure consistency and clarity.

26. P6 L213-L218: make the sentences clear. However, due to the similarity in fuel types between burning spruce and pine logs, as well as spruce and pine branches and needles, they were categorized as separate fuel sources for this test and not compared with each other but were only compared with the other four types of fuels.
- It would be good to make it simple: for this test, 'spruce and pine logs' and 'spruce/pine branches and needles' were in the same fuel type category due to similar characteristics.
- And then readers can have questions, if so (they have same characteristics), why did authors use all of them as different fuels at the beginning?
- If author define 6 different fuel type, please keep the term of 'fuel type' in the whole manuscript. Please don't change to fuel sources or other. It makes confusions. Using term should be clear and same. For burning type as well.
- The sentence below is difficult as well. Similarly, due to the composition of cow dung "cakes," which comprise a mixture of dried cow dung and crop residues and a relatively high correlation between cow dung and straw (Figure 1), the Mann-Whitney U test was carried out without accounting for the presence of the other fuels.

**Response:** Spruce/pine logs and spruce/pine branches and needles were not in the same solid fuel type category. The chemical components produced from burning spruce/pine logs differ significantly from those produced by burning spruce/pine branches and needles. However, in the Mann-Whitney U test, to identify potential markers for one type of fuel, such as spruce/pine logs, the spruce/pine branches and needles were included in the comparison with other fuels. This could result in the loss of many common characteristic markers since these two types of fuel actually come from the same type of tree. Therefore, when identifying markers for spruce/pine logs using the Mann-Whitney U test, spruce/pine branches and needles were not included in the comparison group. Similarly, due to the composition of cow dung "cakes", which are a mixture of dried cow dung and crop residues, the approach used in the Mann-Whitney U test is consistent with the above method.

We have ensured that the term "solid fuel type" is consistently used throughout the manuscript to avoid confusion. We have made sure not to use "fuel sources" or any other terms interchangeably. Similarly, we have maintained

consistent terminology for "burning type" throughout the text. This should provide clarity and prevent any confusion (Line 240-250).

To identify potential markers the Mann-Whitney U test was used to compare the emissions observed for one type of fuel, (e.g., spruce/pine logs) with the gaseous emissions observed for other fuels. The data used for the comparison was the average composition measured throughout a full burning cycle, excluding the initial ignition period. However, due to the similarity in solid fuel types between burning spruce/pine logs, as well as spruce/pine branches and needles, they were categorized as separate solid fuel types for this test and not compared with each other but were only compared with the other four types of fuels. This could result in the loss of many same markers since these two types of fuel actually come from the same type of tree. Therefore, when identifying markers for spruce/pine logs using the Mann-Whitney U test, spruce/pine branches and needles were not included in the comparison group. Similarly, due to the composition of cow dung 'cakes,' which are a mixture of dried cow dung and crop residues, the approach used in the Mann-Whitney U test is consistent with the above method.

27. P7 L226: 'Emission factors from solid-fuel combustion' to 'The characteristics of EF and MCE from different fuel types.' ? make it more clear.

**Response:** We change it to **"**The characteristics of EF and MCE from different solid fuel types".

28. P7 L237: did author mention that NMOG is primary organic gases in the above manuscript? It would make easier to readers if authors mention that NMOG is primary organic gases in the introduction. or Just use NMOG in whole manuscript. It is very important that using same term in whole manuscript for readers.

**Response:** We have reviewed the manuscript and confirmed that we use the term "organic vapors" consistently throughout, aligning with the manuscript's title.

29. P7 L243: The slightly higher EFs for pine and spruce wood burning can be attributed to the more extensive analysis of NMOGs in our study compared to previous research (37.3 245 g/kg) (Hatch et al., 2017).
- Can we say this value is higher than pervious study even though the difference is in the uncertainty levels?

**Response:** Based the comments, we have rephrased this sentence (Line 268-270).

This value is higher than pervious study (37.3 g kg$^{-1}$) even though the difference is in the uncertainty levels , which can be attributed to the more extensive analysis of organic vapor in our study (Hatch et al., 2017).

30. P7 L249: emissions. Does it mean EF? Authors know the difference between emissions and EF.

**Response:** Yes, we corrected it to "EFs" in the manuscript.

31. P7 L250: emission factors. If authors started using EF in place of emission factors, please use EF.

**Response:** We have standardized the terminology throughout the manuscript. We ensured that "EFs" is used

consistently in place of "emission factors" throughout the manuscript.

32. P7 L253: title. Authors never mentioned about primary organic vapors in previous sections. It can be 'The characteristics of NMOG chemical composition from different fuel types'.

**Response:** Based the comments, we revised it to:

3.3 The characteristics of organic vapor from different solid fuel types

33. P7 L254: I don't think section 3.2.1 can be overview. In this section, general characteristics can be handled.

**Response:** Based the comments, we revised it to:

3.3.1 Chemical composition of organic vapor from combustion

34. P7 L255: combustion fuel source mean burning type? Please keep same term.

**Response:** We have ensured that the term "solid fuel types" is consistently used throughout the manuscript to avoid confusion.

35. P7 L257: what is primary NMOGs? So is it different from just NMOGs?

**Response:** We have ensured that the term "organic vapors" is consistently used throughout the manuscript to avoid confusion.

36. P7 L261: biomass samples. What kind of burning type are belonging to biomass samples. Please defined clearly.

**Response:** We have revised it to "solid fuel types" to avoid confusion.

37. P8 L273: What is H/C ratio?

38. P8 L274: What is O/C ratios?

**Response:** We have clarified the definition of H/C and O/C in the manuscript (Line 335-338).

Hydrogen to carbon ratios (H/C, calculated as the ratio of hydrogen atoms to carbon atoms in a molecules).

Oxygen to carbon ratios (O/C, calculated as the ratio of oxygen atoms to carbon atoms in a molecule).

39. P8 L278: Suggested its value. Science paper should suggest certain number otherwise 'slightly lower' means very subjective.

**Response:** We have rephrased this sentence and added the value (Line 340-343).

The results show similarities to the comparison between burning wood and cow dung in the particle phase (Zhang et al., 2023). Specifically, cow dung exhibits a lower fraction of high O/C (0.22) compared to other fuels studied.

40. P8 L290-L301: I suggest this paragraph can come up the beginning of this section. And then authors can easily explain Figure 1 and Figure 2. On the other hand, I also wondered NMOG will be categorized only primary organic gases and secondary as well.

**Response:** In this study, all emissions from solid fuel combustion are considered "primary". To avoid confusion, we have consistently used the term "organic vapors" throughout the manuscript without emphasizing "primary". Additionally, we have moved this explanation to the beginning of the section for better clarity.

41. P9 L303: Hard to understand this sentence. Based on the log10C* values of all organic compounds parameterized with the modified approach of Li et al. (2016) described in Sect. 2.4.

**Response:** We have rephrased this sentence (Line 73-78).

The parameterization described in Sect. 2.4 uses the modified approach of Li et al. (2016) to estimate the volatility of each of the measured compounds by the VOCUS in $\log_{10}(C*)$ [$\mu$g m$^{-3}$]. The gaseous organic compounds were grouped into a 14-bin volatility basis set (VBS) (Donahue et al., 2006) (Figure 4). Following the suggestions in recent papers (Wang et al., 2024; Li et al., 2023; Donahue et al., 2012; Huang et al., 2021; Schervish and Donahue, 2020), the volatility was aggregated into four main classes with units of $\mu$g m$^{-3}$: VOCs as $\log_{10}(C*) > 6.5$, IVOCs as $\log_{10}(C*)$ between 6.5 to 2.5, semi-VOCs (SVOCs) as $\log_{10}(C*)$ between 2.5 to - 0.5 and low-VOCs (LVOCs) as $\log_{10}(C*) < -0.5$).

42. P9 L308: What kind of burning type are involved in biomass burning emissions?

**Response:** We have rephrased this sentence (Line 380-381).

Comparison and compilation of organic vapors sorted by volatility and functional group classification are shown in Figure 4.

43. P9 L311: For all burns. Did author mean for all experiment or for all burning types?

**Response:** Yes. We have corrected it to "all burning types".

44. P9 L320-323: I think the reference cannot support author's finding. In general, spruce and pine branches and needles open burning released a higher proportion of IVOCs (39.3%) into the gas phase compared to stove logs burning (12.6% and 23.9%). Pallozzi et al. (2018) also reported a similar result, showing that needle/leaf combustion released a greater amount of volatile organic compounds into the atmosphere than branch combustion.

**Response:** We have deleted this sentence after carefully reviewing Pallozzi et al. (2018). Upon further examination, we found that the study only mentions specific OVOC compounds being higher in branches and needles, such as the combustion of leaves releasing more benzene than needles, and branches and needle litter

from pine releasing higher amounts than oak. However, this does not sufficiently support our findings, so we have removed the reference Pallozzi et al. (2018).

45. P9 L332: VBS. What does VBS mean?

**Response:** We have added the definition here (Line 73-379).

The parameterization described in Sect. 2.4 uses the modified approach of Li et al. (2016) to estimate the volatility of each of the measured compounds by the VOCUS in $\log_{10}(C^*)$ [μg m$^{-3}$]. The gaseous organic compounds were grouped into a 14-bin volatility basis set (VBS) (Donahue et al., 2006) (Figure 4). Following the suggestions in recent papers (Wang et al., 2024; Li et al., 2023; Donahue et al., 2012; Huang et al., 2021; Schervish and Donahue, 2020), the volatility was aggregated into four main classes with units of μg m$^{-3}$: VOCs as $\log_{10}(C^*) > 6.5$, IVOCs as $\log_{10}(C^*)$ between 6.5 to 2.5, semi-VOCs (SVOCs) as $\log_{10}(C^*)$ between 2.5 to - 0.5 and low-VOCs (LVOCs) as $\log_{10}(C^*) < - 0.5$).

46. P9 L333: wood burning. Did author define wood burning and biomass burning somewhere? What kind of burning type are in this category? And why do authors want to compare two stages? why do authors only analyze wood burning for this comparison?

**Response:** In this study, solid fuels include the biofuels (beech logs, spruce/pine logs, spruce/pine branches and needles, straw, cow dung) and coal. We have revised the sentence: We have added the classification of biomass fuels and coal in the Section 2.1 (Line 118-119). We observed that straw burning typically occurs very rapidly, often within just 10 minutes. While we also compared the flaming and smoldering phases of cow dung burning, the differences between these phases were not significant. Therefore, we focused on wood burning as a representative example to analyze the variations in organic vapor and other particle-phase concentrations, chemical compositions, and other parameters across different combustion stages.

Previous research have demonstrated distinct emission profiles of organic aerosol between flaming and smoldering phases (Heringa et al., 2011; Li et al., 2021). This prompted us to explore the changes in chemical composition and emission factors (EFs) of organic vapors during these different phases. Understanding these variations can provide valuable insights for future research on secondary organic aerosol (SOA) formation and model simulations (Stefenelli et al., 2019).

Six solid fuels were studied (coal briquettes and biomass fuels: beech logs, spruce/pine logs, spruce/pine branches and needles, straw, cow dung) with three to six replicate burns.

47. P9 L335-337: I don't agree with authors point of view. CO increased when the stage was changed from flaming to smoldering. In the top panel, the MCE is used to indicate the flaming stage with a significant CO2 enhancement, while the smoldering stage exhibits high levels of CO.

**Response:** I'm not entirely certain I understood your comment fully, but to avoid any potential misunderstanding,

I have rephrased the sentence for clarity (Line 281-283).

In the top panel, the MCE, CO, and $CO_2$ concentrations, along with our experimental records, are used to indicate the flaming and smoldering stages.

48. P9 L338: What AAE means?

**Response:** "AAE" is an abbreviation for "absorption Ångström exponent" We have added this definition in the manuscript for clarity.

49. P9 L339: f60. Have ever author described what it is in the manuscript or table?

**Response:** "$f60$" is an abbreviation for "the fraction of $m/z$ 60". We have added this definition in the Section 2.2 in the manuscript for clarity (Line 166-168).

In the AMS mass spectra, the fraction of $m/z$ 60 ($f60$) represents the ratio of levoglucosan-like species (Schneider et al., 2006; Alfarra et al., 2007).

50. P9 L341: BB. What does it mean?

**Response:** "BB" is an abbreviation for "biomass burning". We have added this definition earlier in the manuscript for clarity.

51. P10 L350: (31.4 g/kg) and (121.9 g/kg). Please add standard deviation or uncertainty ± value.

**Response:** We have added the standard deviation in the manuscript (Line 294-296).

On average, EFs for organic vapors in the flaming stage are approximately four times lower ($31.4 \pm 7.1$ g kg$^{-1}$) than those in the smoldering stage fires ($121.9 \pm 24$ g kg$^{-1}$).

52. P10 L351: Large similarity. I cannot understand large similarities. Did authors mean similar trend?

**Response:** We acknowledge that the term "large similarity" might be unclear. To clarify, we mean that while there are significant differences in emission strength (such as emission factors or concentrations) between the flaming and smoldering phases, the average carbon and oxygen distribution of organic vapors remains relatively consistent across these phases. In other words, although the overall emission levels vary, the average carbon and oxygen distribution of organic vapors in the emissions does not change significantly (Line 296-298).

Despite significant variability in the strength of organic vapors emissions (EFs), the average carbon and oxygen distribution of organic vapors remained largely consistent across the combustion phases (Figure S4).

53. P10 L352: Figure S7. doesn't have information that what each of #c and #o represents for and also its explanation does not match to the Figure.

**Response:** We apologize for the mistake. The legends for Figure S7 and Figure S8 (now Figure S4 and Figure S5) were mistakenly swapped. We have now corrected this issue so that the figures and their explanations match accurately.

54. P10 L356: What does OVOC stand for?

**Response:** "OVOC" is an abbreviation for "oxygenated VOCs". We have added this definition in the manuscript for clarity.

55. P10 L356: Figure S8. I assumed Figure S8 was not the figure author would like to show.

**Response:** We apologize for the mistake. The legends for Figure S7 and Figure S8 were mistakenly swapped. We have now corrected this issue so that the figures and their explanations match accurately.

56. P10 L365: What does 'common' mean?

**Response:** We have changed the title of this section to "Chemical characteristics of dominant compounds from all biomass fuels". In this section, we aim to distinguish dominate compounds that apply to all biomass fuels (not just a specific type of biomass fuel) and those that are specific to particular solid fuels. The "common" means potential markers that apply to all biomass fuels. However, it is not a scientific term. We have deleted this term in the whole manuscript.

57. P10 L382: Unify the term. there is no 'wood' in Figure 1.

**Response:** In this study, solid fuels include the biofuels (beech logs, spruce/pine logs, spruce/pine branches and needles, straw, cow dung) and coal. We have revised the sentence (Line 422-423):
As shown in Figure 2, biomass fuels (logs, branches and needles, straw and cow dung) are different fuels from coal in this study.

58. P11 L395: solid-fuel combustion. This is very confusing that solid fuel, biomass burning, fuel type, wood... all things are very tangled so that hard to understand. Please keep same term and define each term clearly.

**Response:** The title of this section has been changed to "Identification of potential markers for specific solid fuels". In this study, solid fuels include the biofuels (beech logs, spruce/pine logs, spruce/pine branches and needles, straw, cow dung) and coal. Biomass burning is the burning of biomass biofuels. In the section of 3.4.1, we discuss about the chemical characteristics of dominant compounds from all biomass fuels.

59. P11 L398: add number of table.

**Response:** We have uploaded a separate Excel table as the Supplementary Table.

60. P11 L399: Very confusing between characteristic compounds and selected characteristic compounds. What are the differences between two of them?

61. P11 L401: In contrast, compounds from open burning of straw and cow dung Does it different form characteristic compounds and selected characteristic compounds?

**Response:** We have removed the "selected characteristic compounds"

62. P12 L448: solid fuel combustion, including residential burning (beech logs, a mixture of spruce and pine logs, and coal briquettes) and open combustion (spruce and pine branches and needles, straw, and cow dung).

- Author also investigated biomass burning and woods? did authors use biomass burning same to solid fuel?

- The burning type category is very important here. Please use it clearly.

**Response:** In this study, solid fuels include the biofuels (beech logs, spruce/pine logs, spruce/pine branches and needles, straw, cow dung) and coal. We have added the classification of biomass fuels and coal in the Section 2.1 (Line 118-119). We also rephrased the sentence in the conclusion (Line 48-489).

Six solid fuels were studied (coal briquettes and biomass fuels: beech logs, spruce/pine logs, spruce/pine branches and needles, straw, cow dung) with three to six replicate burns.

In this study, we investigated emissions of organic vapors using Vocus during typical solid fuel combustion, including burning of beech logs, spruce/pine logs, spruce/pine branches and needles, straw, and cow dung and coal briquettes.

63. P13 L481: Still have questions of characteristic compounds and the common compounds.

**Response:** We have addressed some similar comments before.

To avoid ambiguity, we have removed all instances of the term "characteristic compounds" from the manuscript. Instead, we now refer to the selected substances as "potential markers" based on the statistical methods used, similar to those employed by Zhang et al. (2023). This change helps in providing a more precise scientific explanation.

Clarification of "Common" Markers: We have clarified that the term "common" refers to potential markers that are applicable across all biomass fuels, rather than a specific type of biomass fuel. However, recognizing that "common" is not a precise scientific term, we have removed it from the manuscript entirely to avoid any misunderstanding.

64. Figures and table.

- Figures label fonts were too small to read.

- Figure labels and explanations in the captions should be same to the explanation in the manuscript.

- Figure 4. What is key aerosols? Figure 4(b) the colour bar of Smouldering and Flaming can be moved to right side of the panel.

**Response:**

- We have increased the label font size for Figures 1-4 to improve readability. Since the manuscript currently only allows uploading images in PDF format, which may result in compression and reduced image resolution, we will upload the figures separately after the review process is completed.

- We have reviewed and ensured that the labels and explanations in the figure captions match the explanations provided in the manuscript.

- In Figure 4, the term "key aerosols" refers to organic matter (OM) and black carbon (BC). We have clarified this in the figure caption. Additionally, we have adjusted Figure 4(b) by moving the color bar for the smoldering and flaming phases to the right side of the panel, as suggested.

[Figure]

**Figure 1.** (A) Temporal profiles of mixing ratios measured by Vocus and evolution of CO, $CO_2$, AAE, $f60$, MCE and key aerosol compositions during burning cycles of beech logs stove burning (B) Geometric mean of the primary EFs for gas-phase species of different functional groups during flaming and smoldering phase, respectively (the flaming and smoldering was separated by the experimental record and calculated MCE). Error bars correspond to the sample geometric standard deviation of the replicates. The square represents the mixing ratio between smoldering and flaming. In this study, the MCE is used to indicate the flaming stage and smoldering and a significant decrease of MAC and $CO_2$ was observed from the flaming phase to the smoldering phase.

65. Reference.

- Please give the space between the references at least.

**Response:** We will ensure that there is adequate spacing between the references in the revised version of the manuscript.

**Reference:**

[revised manuscript text omitted]

---

## Author Comment (AC3)

**Reviewer #3**

Thank you for the opportunity to review this manuscript. The major concern I have on the manuscript is centered on the analysis of the data. These emissions data are multivariate, strictly positive and relative which means they are compositional data and should be analyzed as such. The data can be transformed into log-ratios which places them on the real number line thus enabling application of many familiar statistical tools. References for this approach to the analysis of emissions data are presented in the comments below. Failure to use this approach can result in spurious correlations between the emissions and errors in interpretation of results. Another commonly used technique in emissions analysis and source apportionment is positive matrix factorization (Sekimoto, K., Koss, A. R., Gilman, J. B., Selimovic, V., Coggon, M. M., Zarzana, K. J., Yuan, B., Lerner, B. M., Brown, S. S., Warneke, C., Yokelson, R. J., Roberts, J. M., and de Gouw, J.: High- and low-temperature pyrolysis profiles describe volatile organic compound emissions from western US wildfire fuels, Atmospheric Chemistry and Physics, 18, 9263–9281, https://doi.org/10.5194/acp-18-9263-2018, 2018). This multivariate technique does not consider the relative nature of emissions data composition.

**Response:** We would like to thank the reviewer for the comments and suggestions to improve the current work. We will have the reviewer comments in black, address the comments in blue, and modified sentences in red.

These emissions data are multivariate, strictly positive and relative which means they are compositional data and should be analyzed as such.

**Response:** We do not believe that compositional data analysis provides the correct basis with which to analyze the data because the data in our work presented here is not strictly positive (i.e., there are zero values), which is a requirement of this data analysis (Greenacre, 2021). The results are not all real, positive, values because there are some species are that observed above our limit of detection in some emissions that are not observed in other types. Consequently, we have ruled out this approach.

Another commonly used technique in emissions analysis and source apportionment is positive matrix factorization (Sekimoto, K., Koss, A. R., Gilman, J. B., Selimovic, V., Coggon, M. M., Zarzana, K. J., Yuan, B., Lerner, B. M., Brown, S. S., Warneke, C., Yokelson, R. J., Roberts, J. M., and de Gouw, J.: High- and low-temperature pyrolysis profiles describe volatile organic compound emissions from western US wildfire fuels, Atmospheric Chemistry and Physics, 18, 9263–9281, https://doi.org/10.5194/acp-18-9263-2018, 2018). This multivariate technique does not consider the relative nature of emissions data composition.

**Response:** As a laboratory we are very familiar with positive matrix factorization (PMF), while PMF could be a useful addition when discussing a time series (similar to the citation shown) (Tong et al., 2021; Qi et al., 2020; Qi et al., 2019; Stefenelli et al., 2019; Wang et al., 2020; Wang et al., 2021; Crippa et al., 2013; Tobler et al., 2014; Crippa et al., 2014; Tobler et al., 2014; Tobler et al., 2014; Crippa et al., 2014; Tobler et al., 20

2021; Mohr et al., 2012). Though, we disagree in its implementation when using average measured composition across many discreet fuel types.

1. 81-83 This is a 1 sentence paragraph. Either expand the text or include it in the preceding or subsequent paragraph.

**Response:** We have already merged this single sentence with the preceding paragraph as suggested.

2. 99 Does the VOCUS identify the characteristic compound or does post-sampling analysis by an investigator identify a "characteristic" compound? What is meant by the term "characteristic" compound for a fuel type? **Response:** To avoid ambiguity, we have removed all instances of the term "characteristic compounds" from the manuscript. Instead, we now refer to the selected substances as "potential markers" based on the statistical methods used, similar to those employed by Zhang et al. (2023). This change helps in providing a more precise scientific explanation.

3. 111 Are you measuring the solid fuel combustion emissions (which would be from char) or the emissions produced from the combustion of gaseous products produced by the pyrolysis of solid fuels?

**Response:** The emissions being measured are combination of both: the solid fuel combustion emissions, which would include emissions from the char, and the emissions produced from the combustion of gaseous products generated by the pyrolysis of the solid fuels. The specific contributions of each depend on factors such as the combustion temperature and conditions. Within this study a full cycle of burning commenced, where the VOCUS measured the composition.

For clarification, we have added the following text (Line 124-138):

With those six different fuels, we categorized six burning types for this experiment. 1) beech logs stove, 2) spruce/pine logs stove, 3) spruce/pine branches and needles open, 4) dry straw open, 5) cow dung open and 6) coal stove. We selected these six solid fuels and conducted emissions tests to simulate certain types of burning found in the atmosphere. Among the list above, 1) beech logs stove and 2) spruce/pine logs stove are representative of residential wood burning, which are burned separately in a stove, consistent with the materials used in two previous articles (Bertrand et al., 2017; Bhattu et al., 2019). To represent forest fires or wildfire and agricultural field combustion, 3) a mixture of fresh spruce/pine branches and needles and 4) straw were combusted in an open stainless-steel cylinder (65 cm in diameter and 35 cm in height). Traditional cooking and heating practices in regions like India are represented by 5) cow dung cakes open burning by using half-open stoves (Loebel Roson et al., 2021). Finally, traditional cooking and heating practices in rural regions of developing countries are represented by 6) coal stove burning in a portable cast iron stove purchased from the local market (Liu et al., 2017). Of course, these conditions do not fully accurately represent the conditions found in actual fires, which consistent of a variety of burning species (e.g., trees, underbrush, peat soils, etc...), but represent laboratory burning conditions.

4. 117 Was either proximate or ultimate analysis performed on the fuels? I would expect a significant difference in N in the cow dung compared to the other fuels. If such a difference exists in the unburnt fuel, it would seem that it would translate through the combustion and into the emissions and identification of the characteristic compounds. The fuel composition is also compositional data and should be analyzed accordingly. **Response:** We agree that the fuel composition would be of interest, but unfortunately proximate / ultimate analysis was not performed on the fuels and lied outside of the scope of these studies. Proximate / ultimate analysis is not routinely measured in accompanied emission measurements, though it is certainly useful when available. Similar to other studies that do not have this type of analysis, the composition of the fuels should be reflected in the emissions observed from the fuels.

5. 125 What was used to represent agricultural waste? Was agricultural was straw only? Please clarify the difference between the agricultural waste and the fuels used to simulate "forest fires"? Was there a difference between the fuel arrangement or the burning conditions? I recommend that you don't use these fuels to characterize "forest fires" as there is a wide range of fuels which burn in forest and bush fires ranging from peat soils to coniferous and hardwood forest fuels to grasses to various shrub fuels.

**Response:** We selected these six solid fuels and conducted emissions tests with different combustion methods to simulate certain types of biomass burning found in the atmosphere. Our goal was not to comprehensively characterize any specific type of combustion, such as forest fires. As the reviewer points out, we did not attempt to replicate the diverse fuel types and conditions present in actual forest fires, which indeed can vary significantly. Instead, we focused on the direct emissions from the selected fuels under controlled conditions. Therefore, we chose a representative fuel, e.g., straw, to test a specific type of agricultural waste. In our subsequent analysis, we focused only on individual fuels like straw, rather than analyzing agricultural waste as a broader category. Accordingly, we have revised the sentences and added explanations to clarify this point (Line 124-138):

With those six different fuels, we categorized six burning types for this experiment. 1) beech logs stove, 2) spruce/pine logs stove, 3) spruce/pine branches and needles open, 4) dry straw open, 5) cow dung open and 6) coal stove. We selected these six solid fuels and conducted emissions tests to simulate certain types of burning found in the atmosphere. Among the list above, 1) beech logs stove and 2) spruce/pine logs stove are representative of residential wood burning, which are burned separately in a stove, consistent with the materials used in two previous articles (Bertrand et al., 2017; Bhattu et al., 2019). To represent forest fires or wildfire and agricultural field combustion, 3) a mixture of fresh spruce/pine branches and needles and 4) straw were combusted in an open stainless-steel cylinder (65 cm in diameter and 35 cm in height). Traditional cooking and heating practices in regions like India are represented by 5) cow dung cakes open burning by using half-open stoves (Loebel Roson et al., 2021). Finally, traditional cooking and heating practices in rural regions of developing countries are represented by 6) coal stove burning in a portable cast iron stove purchased from the local market (Liu et al., 2017). Of course, these conditions do not fully accurately represent the conditions found in actual fires, which consistent of a variety of burning species (e.g., trees, underbrush,

peat soils, etc...), but represent laboratory burning conditions.

6. 133-143 The burning of the logs is described. How did this differ from the straw burning? Straw will ignite and burn more quickly than wooden logs. What was the moisture content of the various fuels? Was a constant heating rate used? These pyrolysis and combustion characteristics will affect time to ignition as well as the composition of the emissions.

**Response:** The method of burning wood (logs) and straw differs, as described in Section 2.1 "Fuel and Burning Types." Straw was combusted in an open stainless-steel cylinder (65 cm in diameter and 35 cm in height), while wood (logs) was burned in a stove. We did measure the moisture content of the woods, where the water content for dried logs was 10-12%, and the water content for the wet (open burning) logs was 30-40%.

We did not use a constant heating rate. Instead, we initiated the burning and then allowed the combustion to proceed according to the properties of the fuels. As expected, we observed that the straw burned faster (fully being consumed within ~3-5 min.) than the logs (burning for ~30-45 min.). In this study, we did not use a heating device to sustain combustion; rather, we aimed to simulate real-world burning conditions, where the fuel burns on its own after ignition. We specifically described the burning of spruce/pine logs in this section because we altered the oxygen content in the stove during the combustion process to explore the changes in emission factors and chemical compositions under different combustion phases (flaming and smoldering) as discussed in Section 3.2. This is particularly relevant since both combustion states are commonly present in household wood burning. Based on your comments, we have revised the sentences and added explanations to clarify this point.

Line 118-119: Six solid fuels were studied (coal briquettes and biomass fuels: beech logs, spruce/pine logs, fresh spruce/pine branches and needles, dry straw, cow dung) with three to six replicate burns.

Line 140-156: The experimental design is shown in Figure S1. In summary, it is made up of a burner and a set of diluters with heated lines. The zero air was provided by a zero air generator (737-250 series, AADCO Instruments, Inc., USA) for cleaning and dilution (Heringa et al., 2011; Bruns et al., 2015). The zero air generator takes ambient air and scrubs particulates and volatile organic compounds from the air leaving a mixture that is largely made up of  $N_2$ ,  $O_2$ , and Ar at ambient concentrations. Other trace gases are scrubbed to lower than atmospheric concentrations including  $CO_2$  (< 80 ppb) and  $CH_4$  (< 40 ppb). Before each burn, a continuous stream of zero air was passed through the gas lines overnight to avoid cross-contamination between burns and to ensure a low background of VOCs. Once a burn is initiated from the various combustibles, emissions are sampled from the chimney through a heated line (473 K). The emissions (both gas and particle phases) are then diluted by two Dekati diluters (DI-1000, Dekati Ltd.) which dilutes the emissions by a factor of ~ 100 (473 K, DI-1000, Dekati Ltd.). Note that beech logs combustion cycles consist of a first cycle referred to as the 'first load' and subsequent cycles, referred to as 'reloads'. The first load consisted of a cold start, flaming, smoldering, and burn-out phase, and the reloads were comprised of a warm start, flaming, smoldering, and burn-out phase. Organic vapor emissions of solid fuel combustion are released within 10-30 min after loading according to the properties of the fuels. We define the time until full ignition duration for burning encompasses 80% of the entire process, starting from loading the fuels to burnout.

Line 357-361: Also, we note a specific difference in the oxygenated aromatic compounds and those with C > 6 for

open wood burning conditions, compared to the stove. This difference may be driven by the difference in the water content of the wood, which is significantly higher for open wood burning (30-40%) compared to stove burning (10-12%). The increase in these oxygenated components comes at the expense of species containing carbonyl and furan functionalities.

7. 180-181 While Andreae and Merlet used the carbon mass balance approach in 2001, it was first used as early as 1969 (Boubel, R. W., Darley, E. F., and Schuck, E. A.: Emissions from burning grass stubble and straw, Journal of the Air Pollution Control Association, 19, 497–500, https://doi.org/10.1080/00022470.1969.10466517, 1969) and was well-established by the mid-1980s (Nelson, R. M., Jr.: An evaluation of the carbon balance technique for estimating emission factors and fuel consumption in forest fire, USDA Forest Service, Southeastern Forest Experiment Station, Asheville, NC, 1982.)

**Response:** Based on the comments from Meinrat O. Andreae and your suggestions, I have updated the references accordingly.

8. 193 Please make sure that the subscripts for C, O, and N are consistently italicized (or not). Make sure that the subscript for oxygen is O and not zero.

**Response:** Thank you for pointing that out. We will ensure that the subscripts for C, O, and N are consistently italicized throughout the text, and we will verify that the subscript for oxygen is correctly represented as "O" and not zero.

$$log_{10_{C}}^{*} = (n_{C}^{0} - n_{C}^{i})b_{C} - n_{O}^{i}b_{O} - 2\frac{n_{C}^{i}n_{O}^{i}}{n_{C}^{i} + n_{O}^{i}}b_{CO} - n_{N}^{i}b_{N}$$
 Equation (3)

9. 202 It has been recently shown that smoke emissions data are multivariate, not independent and are relative values that are dependent on the compounds present in the mixture (Gibergans-Baguena, J., Hervada-Sala, C., and Jarauta-Bragulat, E.: The quality of urban air in Barcelona: a new approach applying compositional data analysis methods, Emerg Sci J, 4, 113–121, https://doi.org/10.28991/esj-2020-01215, 2020; Jarauta-Bragulat, E., Hervada-Sala, C., and Egozcue, J. J.: Air Quality Index revisited from a compositional point of view, Math Geosci, 48, 581–593, https://doi.org/10.1007/s11004-015-9599-5, 2016; Weise, D. R., Palarea-Albaladejo, J., Johnson, T. J., and Jung, H.: Analyzing wildland fire smoke emissions data using compositional data techniques, J. Geophys. Res. Atmos., 125, e2019JD032128, https://doi.org/10.1029/2019JD032128, 2020). These characteristics of the data apply whether the emissions data are expressed as emission factors, emission ratios, mole ratios or mass ratios (van den Boogaart, K. G. and Tolosana-Delgado, R.: Analyzing compositional data with R, Springer, Heidelberg, 258 pp., 2013.).

It has also been shown that MCE as an index describing the completeness of combustion is not independent of

the quantities of other emissions and that it should not be used as a predictor for the other gases in the composition. **Response:** This study presents the emissions from many different fuel sources and demonstrates that the composition of the gaseous emissions is indeed dependent upon the compounds present within the fuel mixture itself (assuming the composition of the emissions are similar to those in the combustion source). We demonstrate that changing the MCE impacts the emission factors and the concentrations of the emitted gases within the measurement itself and specifically changes the composition of the emissions as well. This work also demonstrates there is a specific change in the chemical composition of the emissions when the MCE changes.

10. How is the Mann-Whitney test and other techniques used in this manuscript affected by the statistical characteristics of your data? The Mann-Whitney test is a univariate test. You should consider using the generalized multivariate version if it has been applied to compositional data (https://doi.org/10.1016/j.jmva.2022.104946). As compositional data analysis has been used more extensively in Europe, recommend reaching out to the statisticians listed in the various publications above. You should also consider a global test (instead of pairwise comparisons) that controls the experiment-wise probability of committing a Type 1 error (such as false discovery rate-Benjamini, Y. and Hochberg, Y.: Controlling the false discovery rate: a practical and powerful approach to multiple testing, Journal of the Royal Statistical Society. Series B (Methodological), 57, 289-300, 1995).

**Response:** Although it is possible to use a multi-variate approach, we have chosen to use Mann-Whitney to identify outliers (potential markers) to be consistent with previous work and align with identification of markers used in various other mass spectrometric approaches in analytical chemistry (White et al., 2019; Chen et al., 2012; Teunissen et al., 2011; Chmaj-Wierzchowska et al., 2015; Nomura et al., 2004; Jasperse et al., 2007; Nagai et al., 2020; Sun et al., 2019; Tritten et al., 2013). Further the data that is being used is not always positive (there are 0 values), which means the use of compositional data analysis is not a valid approach for our data (Greenacre, 2021). We have chosen the univariate approach to identify molecular formula that are specific statistical outliers relative to the other emissions, which differs from other studies that have focused on emission factors from specific sources. Only species that are confirmed outliers between each and every pair-wise comparison is chosen, we believe this is a cautious approach that lowers the probability of committing a Type 1 error is 5% and 4 groups are used, then the probability of committing a Type 1 error across 4 different comparisons is 0.00062%.

11. 232 What does 0.99 +/- 0.02 mean? Is this arithmetic mean and standard error or standard deviation? Since MCE is a proportion that can not exceed 1, the correct formula for the confidence interval of this proportion should not exceed 1. The geometric mean is the appropriate measure of central tendency for relative (proportional

data). EFs are expressed as gm pollutant/gm fuel burned which is a rate (and a relative value so a geometric mean should be used as in Butler, B. M., Palarea-Albaladejo, J., Shepherd, K. D., Nyambura, K. M., Towett, E. K., Sila, A. M., and Hillier, S.: Mineral–nutrient relationships in African soils assessed using cluster analysis of X-ray powder diffraction patterns and compositional methods, Geoderma, 375, 114474, https://doi.org/10.1016/j.geoderma.2020.114474, 2020.

**Response:** We apologize for the error in reporting  $0.99 \pm 0.02$ . Based on the data from Table S1, the standard deviation is actually less than 0.001, so we have corrected this and omitted the unnecessary margin. In this study, the emission factors (EFs) are calculated as arithmetic mean  $\pm$  standard deviation. We chose this method to maintain consistency with previous studies that calculated VOC, CO and CO2 emission factors using the same approach. This allows for direct comparison with their results (Andreae, 2019; Janhäll et al., 2010; Koss et al., 2018).

12. 257 Correlation is not an appropriate measure for compositional data as the value of the correlation coefficient is dependent upon the other compounds in the composition. Dropping a gas from the composition changes the pairwise correlations (Aitchison, J.: A concise guide to compositional data analysis, 2003. Available at http://ima.udg.edu/activitats/codawork03/; Weise, D. R., Fletcher, T. H., Safdari, M.-S., Amini, E., and Palarea-Albaladejo, J.: Application of compositional data analysis to determine the effects of heating mode, moisture status and plant species on pyrolysates, Int. J. Wildland Fire, 31, 24–45, https://doi.org/10.1071/WF20126, 2022). Proportionality has been suggested as an appropriate measure of the association between two components of a composition (Lovell, D., Pawlowsky-Glahn, V., Egozcue, J. J., Marguerat, S., and Bähler, J.: Proportionality: a valid alternative to correlation for relative data, PLoS Comput Biol, 11, e1004075, https://doi.org/10.1371/journal.pcbi.1004075, 2015). Drawing conclusions based on the measure of association between two variables without determining the significance of the measure by a statistical test of some sort is not recommended.

**Response:** We agree with the reviewer that a simple correlation matrix is likely not the most appropriate measure for compositional data if we were trying to demonstrate anything quantitative. The reviewer should also consider the point of using such a simplistic approach as a tool to discuss the results. The use of a correlation matrix as shown in Figure 2 highlights the similarity between all biomass combustion sources investigated, and that there is a specific difference for coal. The correlation matrix is not used in absolute terms regarding the emissions but is used in qualitative terms as a discussion tool in order to discuss the reproducibility from burn to burn and the similarity (or differences) between the emissions which are shown in detail in Figures 4 and 5, where a specific difference for coal comes from the emissions of non-oxygenated aromatic compounds. It would be just as feasible to put 6 mass spectra representing the different emission types.

Note the use of the correlation matrix on lines 316 - 324:

To assess the feasibility of distinguishing differences between combustion solid fuel types based on the measured species, we evaluated the similarity of the mass spectra obtained from each experiment using the correlation coefficient (r), as shown in Figure 2. Organic vapors from the same burning fuel are strongly correlated (0.82-0.99), indicating the general repeatability of the experiments. Furthermore, we observed a weak intra-fuel correlation between coal and other biomass sources (0.44-0.78), suggesting significant differences in chemical composition. By contrast, the separation between different solid fuel type is not stark and all possess a correlation between 0.6-0.98. Overall, the correlation coefficient highlights similarities between all biomass-based emissions, which will now be discussed in detail.

13. 290 In compositional data analysis, the effect of fuel type on the log-ratios between different groups of compounds can be tested in an analysis of variance context to determine differences. These log-ratios are known as balances (Egozcue, J. J. and Pawlowsky-Glahn, V.: Groups of parts and their balances in compositional data analysis, Mathematical Geology, 37, 795–828, https://doi.org/10.1007/s11004-005-7381-9, 2005; Weise, D. R., Fletcher, T. H., Safdari, M.-S., Amini, E., and Palarea-Albaladejo, J.: Application of compositional data analysis to determine the effects of heating mode, moisture status and plant species on pyrolysates, Int. J. Wildland Fire, 31, 24–45, https://doi.org/10.1071/WF20126, 2022).

**Response:** As stated above, we do not believe that compositional data analysis provides the correct basis with which to analyze the data because the data in our work presented here is not strictly positive (i.e., there are zero values), which is a requirement of this data analysis (Greenacre, 2021). The results are not all real, positive, values because there are some species are that observed above our limit of detection in some emissions that are not observed in other types. Consequently, we have ruled out this approach.

14. 308 see comment below for figures S4, S5 regarding error bars.

**Response:** We have double-checked the data and corrected that the error bar represents 1 standard deviation, not 1/2. We have corrected them in the SI.

15. 348 You are discussing differences in relative terms which is appropriate. The statistics used to describe and test hypotheses should also recognize the relative nature of the data.

**Response:** I agree that addressing the relative nature of the data is essential, especially when making comparisons. I'll ensure that the statistical methods used align with this approach and clearly reflect the relative differences within the data. We have added the standard deviation in the manuscript (Line 294-296).

On average, EFs for organic vapors in the flaming stage are approximately four times lower  $(31.4 \pm 7.1 \text{ g kg}^{-1})$  than those in the smoldering stage fires  $(121.9 \pm 24 \text{ g kg}^{-1})$ .

16. 370 See the earlier comment regarding the Mann-Whitney test, the multivariate nature of the data and the probability of committing a Type 1 error.

**Response:** We have chosen the univariate approach to identify molecular formula that are specific statistical outliers relative to the other emissions, which differs from other studies that have focused on emission factors from specific sources. Only species that are confirmed outliers between each and every pair-wise comparison is chosen, we believe this is a cautious approach that lowers the probability of committing a Type 1 error, which is already low in the case for a single pair-wise comparison. If the probability of committing a Type 1 error is 5% and 4 groups are used, then the probability of committing a Type 1 error across 4 different comparisons is 0.00062%.

17. 398 Which supplementary table contains the characteristics compounds?**Response:** We realized that we forgot to upload the Excel file. The file has now been uploaded.

**18. 423 Where is the chemical composition of unburnt cow dung presented?**

Table 1 Are the values arithmetic mean +/- standard deviation? Please provide more information on values. Should use geometric mean and present a confidence interval (or the standard error of the mean). Consider including complete fuel composition (CHNSO). Also, proximate analysis because cows have ability to digest cellulose which make affect the burning characteristics or the relative amounts of cellulose, hemicellulose and lignin in the fuels which will affect both the pyrolysis and combustion processes as well as emissions production. **Response:** As mentioned above, unburnt chemical analysis of the cow dung was not performed.

**19. Figure 2-5 Recommend making the axes and other information larger fonts (relative to titles).**

**Response:** Thank you for your suggestion. We have updated figures to increase the font size of the axes and other information relative to the titles to improve readability.

**20. Figure 5 What do the different sized circles labeled 0.1, 0.5, 1 and 2 indicate?**

**Response:** In the caption, we mentioned that the markers are scaled according to the square root of the fractional contribution (%). Therefore, the sizes of the circles labeled 0.1, 0.5, 1, and 2 represent the square roots of the corresponding fractional contributions.

21. Figure S4, S5 Why is ½ of 1 standard deviation used as an error bar? Is this is based on the assumption made for normally distributed data that 1 standard deviation captures about 68 percent of the data and 2 standard deviations capture about 95% of the data? This is based on the population and not the sample. A confidence

interval should be calculated. Also, these data are not normally-distributed. They are proportions which are constrained between 0 and 1 (or 0 and 100).

**Response:** We have double-checked the data and corrected that the error bar represents 1 standard deviation, not ½. While the data from a single experiment may not be normally distributed, the average data and standard deviation were calculated from multiple repeated experiments. The error bars reflect the standard deviation of the relative contributions across these repeated experiments.

**22. Figure S13 Caption needs to be fixed.**

**Response:** We have revised the caption.

**Reference:**

- Andreae, M. O.: Emission of trace gases and aerosols from biomass burning-an updated assessment, Atmos. Chem. Phys., 19, 8523-8546, 2019.
- Bertrand, A., Stefenelli, G., Bruns, E. A., Pieber, S. M., Temime-Roussel, B., Slowik, J. G., Prévôt, A. S. H., Wortham,
  H., El Haddad, I., and Marchand, N.: Primary emissions and secondary aerosol production potential from woodstoves for residential heating: Influence of the stove technology and combustion efficiency, Atmos. Environ., 169, 65-79, 10.1016/j.atmosenv.2017.09.005, 2017.
- Bhattu, D., Zotter, P., Zhou, J., Stefenelli, G., Klein, F., Bertrand, A., Temime-Roussel, B., Marchand, N., Slowik, J.
  G., Baltensperger, U., Prevot, A. S. H., Nussbaumer, T., El Haddad, I., and Dommen, J.: Effect of Stove Technology and Combustion Conditions on Gas and Particulate Emissions from Residential Biomass Combustion, Environ. Sci. Technol., 53, 2209-2219, 10.1021/acs.est.8b05020, 2019.
- Bruns, E. A., El Haddad, I., Keller, A., Klein, F., Kumar, N. K., Pieber, S. M., Corbin, J. C., Slowik, J. G., Brune, W. H., Baltensperger, U., and Prévôt, A. S. H.: Inter-comparison of laboratory smog chamber and flow reactor systems on organic aerosol yield and composition, Atmos. Meas. Tech., 8, 2315-2332, 10.5194/amt-8-2315-2015, 2015.
- Chen, Y.-T., Chen, H.-W., Domanski, D., Smith, D. S., Liang, K.-H., Wu, C.-C., Chen, C.-L., Chung, T., Chen, M.-C., and Chang, Y.-S.: Multiplexed quantification of 63 proteins in human urine by multiple reaction monitoringbased mass spectrometry for discovery of potential bladder cancer biomarkers, Journal of proteomics, 75, 3529-3545, 2012.
- Chmaj-Wierzchowska, K., Kampioni, M., Wilczak, M., Sajdak, S., and Opala, T.: Novel markers in the diagnostics of endometriomas: Urocortin, ghrelin, and leptin or leukocytes, fibrinogen, and CA-125?, Taiwanese Journal of Obstetrics and Gynecology, 54, 126-130, 2015.
- Crippa, M., El Haddad, I., Slowik, J. G., DeCarlo, P. F., Mohr, C., Heringa, M. F., Chirico, R., Marchand, N., Sciare,

J., and Baltensperger, U.: Identification of marine and continental aerosol sources in Paris using high resolution aerosol mass spectrometry, Journal of Geophysical Research: Atmospheres, 118, 1950-1963, 2013.

Greenacre, M.: Compositional data analysis, Annual Review of Statistics and its Application, 8, 271-299, 2021.

- Heringa, M. F., DeCarlo, P. F., Chirico, R., Tritscher, T., Dommen, J., Weingartner, E., Richter, R., Wehrle, G., Prévôt,
  A. S. H., and Baltensperger, U.: Investigations of primary and secondary particulate matter of different wood combustion appliances with a high-resolution time-of-flight aerosol mass spectrometer, Atmos. Chem. Phys., 11, 5945-5957, 10.5194/acp-11-5945-2011, 2011.
- Janhäll, S., Andreae, M. O., and Pöschl, U.: Biomass burning aerosol emissions from vegetation fires: particle number and mass emission factors and size distributions, Atmos. Chem. Phys., 10, 1427-1439, 2010.
- Jasperse, B., Jakobs, C., Eikelenboom, M. J., Dijkstra, C. D., Uitdehaag, B. M., Barkhof, F., Polman, C. H., and Teunissen, C. E.: N-acetylaspartic acid in cerebrospinal fluid of multiple sclerosis patients determined by gaschromatography-mass spectrometry, Journal of neurology, 254, 631-637, 2007.
- Koss, A. R., Sekimoto, K., Gilman, J. B., Selimovic, V., Coggon, M. M., Zarzana, K. J., Yuan, B., Lerner, B. M., Brown, S. S., and Jimenez, J. L.: Non-methane organic gas emissions from biomass burning: identification, quantification, and emission factors from PTR-ToF during the FIREX 2016 laboratory experiment, Atmos. Chem. Phys., 18, 3299-3319, 2018.
- Liu, C., Zhang, C., Mu, Y., Liu, J., and Zhang, Y.: Emission of volatile organic compounds from domestic coal stove with the actual alternation of flaming and smoldering combustion processes, Environ. Pollut., 221, 385-391, 10.1016/j.envpol.2016.11.089, 2017.
- Loebel Roson, M., Duruisseau-Kuntz, R., Wang, M., Klimchuk, K., Abel, R. J., Harynuk, J. J., and Zhao, R.: Chemical Characterization of Emissions Arising from Solid Fuel Combustion—Contrasting Wood and Cow Dung Burning, ACS Earth and Space Chemistry, 5, 2925-2937, 10.1021/acsearthspacechem.1c00268, 2021.
- Mohr, C., DeCarlo, P., Heringa, M., Chirico, R., Slowik, J., Richter, R., Reche, C., Alastuey, A., Querol, X., and Seco, R.: Identification and quantification of organic aerosol from cooking and other sources in Barcelona using aerosol mass spectrometer data, Atmos. Chem. Phys., 12, 1649-1665, 2012.
- Nagai, K., Uranbileg, B., Chen, Z., Fujioka, A., Yamazaki, T., Matsumoto, Y., Tsukamoto, H., Ikeda, H., Yatomi, Y., and Chiba, H.: Identification of novel biomarkers of hepatocellular carcinoma by high-definition mass spectrometry: Ultrahigh-performance liquid chromatography quadrupole time-of-flight mass spectrometry and desorption electrospray ionization mass spectrometry imaging, Rapid Commun. Mass Spectrom., 34, e8551, 2020.
- Nomura, F., Tomonaga, T., Sogawa, K., Ohashi, T., Nezu, M., Sunaga, M., Kondo, N., Iyo, M., Shimada, H., and Ochiai, T.: Identification of novel and downregulated biomarkers for alcoholism by surface enhanced laser desorption/ionization-mass spectrometry, Proteomics, 4, 1187-1194, 2004.

- Qi, L., Chen, M., Stefenelli, G., Pospisilova, V., Tong, Y., Bertrand, A., Hueglin, C., Ge, X., Baltensperger, U., Prévôt, A. S. H., and Slowik, J. G.: Organic aerosol source apportionment in Zurich using an extractive electrospray ionization time-of-flight mass spectrometer (EESI-TOF-MS) Part 2: Biomass burning influences in winter, Atmos. Chem. Phys., 19, 8037-8062, 10.5194/acp-19-8037-2019, 2019.
- Qi, L., Vogel, A. L., Esmaeilirad, S., Cao, L., Zheng, J., Jaffrezo, J.-L., Fermo, P., Kasper-Giebl, A., Daellenbach, K. R., Chen, M., Ge, X., Baltensperger, U., Prévôt, A. S. H., and Slowik, J. G.: A 1-year characterization of organic aerosol composition and sources using an extractive electrospray ionization time-of-flight mass spectrometer (EESI-TOF), Atmos. Chem. Phys., 20, 7875-7893, 10.5194/acp-20-7875-2020, 2020.
- Stefenelli, G., Pospisilova, V., Lopez-Hilfiker, F. D., Daellenbach, K. R., Hüglin, C., Tong, Y., Baltensperger, U., Prévôt, A. S. H., and Slowik, J. G.: Organic aerosol source apportionment in Zurich using an extractive electrospray ionization time-of-flight mass spectrometer (EESI-TOF-MS) – Part 1: Biogenic influences and day– night chemistry in summer, Atmos. Chem. Phys., 19, 14825-14848, 10.5194/acp-19-14825-2019, 2019.
- Sun, Y., Chen, Y., Sun, C., Liu, H., Wang, Y., and Jiang, X.: Analysis of volatile organic compounds from patients and cell lines for the validation of lung cancer biomarkers by proton-transfer-reaction mass spectrometry, Analytical Methods, 11, 3188-3197, 2019.
- Teunissen, C., Koel-Simmelink, M., Pham, T., Knol, J., Khalil, M., Trentini, A., Killestein, J., Nielsen, J., Vrenken, H., and Popescu, V.: Identification of biomarkers for diagnosis and progression of MS by MALDI-TOF mass spectrometry, Multiple Sclerosis Journal, 17, 838-850, 2011.
- Tobler, A. K., Skiba, A., Canonaco, F., Močnik, G., Rai, P., Chen, G., Bartyzel, J., Zimnoch, M., Styszko, K., and Nęcki, J.: Characterization of non-refractory (NR) PM 1 and source apportionment of organic aerosol in Kraków, Poland, Atmos. Chem. Phys., 21, 14893-14906, 2021.
- Tong, Y., Pospisilova, V., Qi, L., Duan, J., Gu, Y., Kumar, V., Rai, P., Stefenelli, G., Wang, L., Wang, Y., Zhong, H., Baltensperger, U., Cao, J., Huang, R.-J., Prévôt, A. S. H., and Slowik, J. G.: Quantification of solid fuel combustion and aqueous chemistry contributions to secondary organic aerosol during wintertime haze events in Beijing, Atmos. Chem. Phys., 21, 9859-9886, 10.5194/acp-21-9859-2021, 2021.
- Tritten, L., Keiser, J., Godejohann, M., Utzinger, J., Vargas, M., Beckonert, O., Holmes, E., and Saric, J.: Metabolic profiling framework for discovery of candidate diagnostic markers of malaria, Scientific reports, 3, 2769, 2013.
- Wang, L., Slowik, J. G., Tong, Y., Duan, J., Gu, Y., Rai, P., Qi, L., Stefenelli, G., Baltensperger, U., Huang, R.-J., Cao, J., and Prévôt, A. S. H.: Characteristics of wintertime VOCs in urban Beijing: Composition and source apportionment, Atmos. Environ., 9, 100100, 10.1016/j.aeaoa.2020.100100, 2021.
- Wang, L., Slowik, J. G., Tripathi, N., Bhattu, D., Rai, P., Kumar, V., Vats, P., Satish, R., Baltensperger, U., Ganguly, D., Rastogi, N., Sahu, L. K., Tripathi, S. N., and Prévôt, A. S. H.: Source characterization of volatile organic compounds measured by proton-transfer-reaction time-of-flight mass spectrometers in Delhi, India, Atmos.

Chem. Phys., 20, 9753-9770, 10.5194/acp-20-9753-2020, 2020.

- White, R., Pulford, E., Elliot, D. J., Thurgood, L. A., and Klebe, S.: Quantitative mass spectrometry to identify protein markers for diagnosis of malignant pleural mesothelioma, Journal of proteomics, 192, 374-382, 2019.
- Zhang, J., Li, K., Wang, T., Gammelsæter, E., Cheung, R. K., Surdu, M., Bogler, S., Bhattu, D., Wang, D. S., and Cui,
  T.: Bulk and molecular-level composition of primary organic aerosol from wood, straw, cow dung, and plastic burning, Atmos. Chem. Phys., 23, 14561-14576, 2023.